# Characterization and ligand binding properties of a fatty acid- and retinol- binding protein (Hp-FAR-2) from *Heligmosomoides polygyrus*

Pakeeza Azizpor[1], Janice Montoya[1], Fayez Eyabi[1], Jose Ramirez[1], Tara Hill[2], Robert Pena[1], Manisha Mishra[1], Martin J. Boulanger[2], Adler R. Dillman[1]*

**1** Department of Nematology, University of California, Riverside, California, United States of America,
**2** Department of Biochemistry and Microbiology, University of Victoria, Victoria, Canada

* adlerd@ucr.edu

## Abstract

Parasitic nematodes are major pathogens of humans, animals, and plants, contributing to global health challenges and substantial agricultural losses. Fatty acid- and retinol-binding proteins (FARs), secreted by parasitic nematodes, are believed to play key roles in host–pathogen interactions, including immune modulation and nutrient acquisition. In this study, we characterize a FAR protein from the gastrointestinal nematode *Heligmosomoides polygyrus*, Hp-FAR-2. Hp-FAR-2, unlike FARs from *Caenorhabditis elegans*, *Steinernema carpocapsae*, and *Ancylostoma ceylanicum*, did not influence immunity or survival in a *Drosophila melanogaster* infection model, suggesting functional divergence within the FAR family. Competitive lipid-binding assays revealed a preference for omega-3 and omega-6 polyunsaturated fatty acids, indicating selective binding to bioactive lipids that may modulate immunity. Using RAW 264.7 macrophages, we found that Hp-FAR-2 suppresses the expression of both M1-associated (TNF-α, IL-6) and M2-associated (Chil3) markers during polarization, implicating it as a broad immunomodulator that may inhibit inflammatory responses and tissue repair mechanisms to promote chronic infection. Our findings support a model in which Hp-FAR-2 disrupts host lipid signaling and immune function to favor parasite persistence, suggesting its potential role in the excretory/secretory products of *H. polygyrus*. These findings enhance our understanding of FAR-mediated host manipulation and may inform the development of novel anthelmintic or immunoregulatory therapies.

### Author summary

Parasitic worms infect billions of people worldwide and cause long-term, chronic infections that remain a major public health challenge. To survive inside their hosts, these parasites release secreted proteins that alter immune responses

**Data availability statement:** All data is available in the supplemental and through Mendeley Data via this link: https://data.mendeley.com/datasets/2bt7xj2ydj/2.

**Funding:** This work was supported by the National Institute of General Medical Sciences of the National Institutes of Health (award no. R35GM137934 to ARD). The funders had no role in study design, data collection and analysis, decision to publish, or preparation of the manuscript.

**Competing interests:** The authors have declared that no competing interests exist.

and help the worms persist. One protein family unique to nematodes is the fatty acid- and retinol-binding proteins (FARs), which are thought to bind host lipids and interfere with normal signaling. In this study, we investigated a previously uncharacterized FAR protein from the intestinal parasite *Heligmosomoides polygyrus*, called Hp-FAR-2. We found that Hp-FAR-2 binds selectively to omega-3 and omega-6 fatty acids, which are important regulators of inflammation and immunity. When tested in fruit flies, Hp-FAR-2 did not affect survival or immune responses, suggesting that its function may be specific to mammalian hosts. In experiments with mouse macrophages, Hp-FAR-2 reduced the expression of genes linked to both inflammation and tissue repair. These findings show that Hp-FAR-2 can target key immune pathways by binding host lipids, which may promote parasite survival. Understanding these mechanisms could help identify new strategies for controlling worm infections or even inspire therapies for inflammatory diseases.

## Introduction

Parasitic nematodes cause diseases in humans, animals, and plants, leading to substantial losses in agriculture and posing significant health challenges to humans. For instance, intestinal nematode infections caused by soil-transmitted helminths (STH), infect more than 1 billion people throughout the world [1]. Parasitic nematodes are constantly exposed to host immune effector mechanisms and have evolved mechanisms to ensure survival in the host [2]. One immune avoidance strategy is the release of a wide range of molecules including nucleic acids, proteins, lipids, and carbohydrates, referred to as the excretory-secretory (ES) products [3]. The ES products of parasitic nematodes are released into the surrounding tissues of the host, which can interfere with host signaling mechanisms and immune homeostasis, aiding parasite survival. Individual molecules from the ES products of various helminths have been found to exhibit immunomodulatory properties [3]. Some of these molecules have been shown to target every phase of the immune response to helminth infection including initiation, antigen recognition, adaptive and effector cell responses, and healing and remodeling [3]. More recently, immunomodulatory molecules derived from the ES products of parasites are being studied as potential therapeutics for inflammatory diseases [4–6].

A distinctive protein family frequently found in the ES products of nematodes is the fatty acid- and retinol-binding proteins (FARs). FARs are alpha-helix rich proteins only found in nematodes, which occur in several isoforms of ~14–20 kDa, that display ligand multiplicity for fatty acids and bind to retinol [7,8]. Many FARs have been found in the hypodermis, cuticle surface region, and esophageal glands of nematodes [9–13]. Furthermore, many FAR proteins have a signal peptide and are found in the ES products of various parasitic nematodes [14–17]. FARs are suggested to function in nutrient acquisition, growth, development, reproduction, and modulation of host immunity during parasitic nematode infection [9,10,18,19]. The function of

FARs highlights the importance of fatty acids in nematode physiology and parasitism. Fatty acids and retinol are essential for both vertebrate and invertebrate physiology. In vertebrates, fatty acids regulate membrane composition and are precursors of powerful lipid signaling molecules [20]. Parasitic nematodes are unable to synthesize fatty acids and retinol de novo and must acquire these lipids from their host or the environment [21]. The role of fatty acids in nematode physiology has been studied using the free-living nematode *Caenorhabditis elegans*, which can synthesize polyunsaturated fatty acids (PUFAs) de novo and contains at least seven desaturases and two elongases as a part of the PUFA synthetic pathway [22,23]. Inactivation of these genes has been shown to affect different nematode physiologic processes such as reduced body size, growth retardation, deviated body adiposity, reproductive defects, changes in physiologic rhythms, slowed movement, reduced adult life span, defects in sensory signaling and neurotransmission, and reproduction at lower temperatures [24–31]. Fatty acids also play an important role in cuticle construction, growth, and development [32]. Retinoids have also been suggested to play a role in parasitic nematode growth and development [33]. Nematodes possess several types of lipid-binding proteins. One example is the nematode polyprotein antigens (NPAs), which are synthesized as large polyprotein precursors that are post-translationally processed down to multiple copies of small lipid binding proteins of ~14 kDa [34]. NPAs bind to fatty acids and retinoids and can elicit a strong host immune response in parasitic nematodes [35]. Another example is fatty acid binding proteins (FABPs), which are found in vertebrates and invertebrates, and exhibit binding affinities for various fatty acids [36]. Finally, FARs are an understudied protein family unique to nematodes with larger ligand-multiplicity with respect to FABPs [7]. They have been shown in vitro to bind to fatty acids and retinol molecules, including those important to immune signaling in mammals [37], making them a compelling target for understanding host-parasite interactions and uncovering novel mechanisms of immune modulation.

Many studies on FARs have been in vitro, studying their structure, ligands, and expression patterns [10,11,14,16,38,39]. Fewer studies have looked at how FARs from parasitic nematodes affect host tissues. Much work has been done to evaluate the roles of FARs in plant-parasite interactions. The first plant-parasitic FAR discovered was Gp-FAR-1, a FAR in the pale cyst nematode, *Globodera pallida,* which was found to bind to precursors of the jasmonic acid signaling pathway and inhibits lipoxygenase activity in the plant host [11]. Jasmonic acid is an important signal transducer in systemic plant immunity [40]. Further evidence that FARs alter susceptibility to infection was shown with Mj-FAR-1, a FAR from the plant-parasitic nematode *Meloidogyne javanica*. Transgenic tomato roots expressing Mj-FAR-1 showed higher susceptibility to nematode infection and allowed for faster nematode growth once infected [12].

Few studies have examined the immunomodulatory effects of FARs in animal-parasitic nematodes. Two FARs from filarial nematode *Burgia malayi,* Bm-FAR-1 and Bm-FAR-2 are targets of strong IgG1, IgG3, and IgE antibody responses in infected individuals [17]. Furthermore, immunization with recombinant Bm-FAR-1 in gerbils conferred significant protection against challenge of third-stage larvae (L3*) B. malayi,* but only when formulated in water-in-oil adjuvant [17]. Two FARs from the insect parasite *Steinernema carpocapsae*, Sc-FAR-1 and Sc-FAR-2, have been shown to modulate *Drosophila melanogaster* immunity. *S. carpocapsae* FARs were shown to dampen the *D. melanogaster* immune system including key components of the fly immune response including the phenoloxidase cascade and antimicrobial peptide production (AMP) [18]. Furthermore, a FAR from the free-living nematode *Caenorhabditis elegans* and the human hookworm *Ancylostoma ceylanicum* dampen fly immunity, decreasing resistance to infection [18]. More recently, a FAR from dog hookworm, *Ancylostoma caninum*, was found to decrease inflammation, decreasing pathology in a mouse model of colitis [41].

Here we characterize a FAR from the murine parasite *Heligmosomoides polygyrus*. *H. polygyrus* is a chronic intestinal parasitic nematode known to modify a wide spectrum of the host immune response, and primary infections can persist for many months in susceptible strains [42,43]. The life cycle is well-defined in experimental settings beginning with L3 larvae ingested via oral gavage which then penetrate through the submucosa of the small intestine within 24 hours. There, L3 larvae undergo two developmental molts before emerging back into the lumen as adult worms, which feed on host intestinal tissue. *H. polygyrus* has been found to exhibit immunomodulatory properties that extend far beyond the site of infection, and this effect is hypothesized to be partially mediated by the secretome, which mediates interactions

with the host through direct contact with host cells in proximity to the worm, and potentially systemically [43]. The host immune response to this infection is a predominantly Th2 response phenotype, where upon secondary infection, memory CD4+ T cells rapidly accumulate at the host-parasite interface and secrete IL-4, inducing localized development of alternatively activated macrophages (AAMacs) recruited to this site. These macrophages, which rely on arginase activity like arginase-1 (Arg1), and secretion of chitinase-like protein 3 (Chil3), and resistin-like molecule alpha (RELM-α), impair parasite health and mobility, aiding in worm expulsion [44–46]. Thus, *H. polygyrus* serves as a relatively stable system in mice to analyze immune regulation in chronic infection [42]. The excreted-secreted products of *H. polygyrus* (HES) serve as the primary interface between host-parasite interaction, exhibiting a broad range of immunomodulatory effects [43]. For example, the HES has been shown to inhibit activation of dendritic cells, induction of Tregs, and block airway allergy in vivo. [47–49]. As such, there is significant interest in identifying the individual components of the HES for the development of potential novel therapies for inflammatory diseases [50]. Multiple studies have defined the various components of the HES and have classified FARs as one of the protein families present [51–54]. A previous study characterized a FAR from *H. polygyrus,* Hp-FAR-1, identified in the fourth larval stage (L4) ES products [16]. Hp-FAR-1 was found to bind to retinol and oleic acid in vitro however no analysis of the in vivo role was explored [16].

Despite the thorough analysis of the individual components of the HES, the role of the FAR protein family in the HES, and in nematode infections in general, is understudied. Here we characterize Hp-FAR-2, another *H. polygyrus* FAR found in the HES [51,53,54], that may play a key role in host-parasite interactions. In vitro assays identified that Hp-FAR-2 binds to various fatty acids and retinol including precursors of the lipid signaling pathway in mammals. Analysis in a *D. melanogaster* infection model indicates that Hp-FAR-2 may be targeting a host-specific pathway not conserved in *Drosophila*. Lastly, we utilized a murine macrophage cell line to examine the role of Hp-FAR-2 in host immunity. We found that Hp-FAR-2 attenuates the expression of tumor necrosis factor-alpha (*TNF-α*) and interleukin-6 (*IL-6*) in M1 macrophages and *Chil3* in M2 macrophages, suggesting an immunoregulatory function within the HES. Together, our data suggest Hp-FAR-2 can modulate host macrophage expression, which may function in the HES to maintain immune homeostasis, minimizing inflammation and prolonging survival of *H. polygyrus* within the host. These findings are a step forward into elucidating the role of Hp-FAR-2 in the HES and they offer new insights into the function of FARs in animal-parasitic nematodes.

## Methods

### Sequence analysis, alignment and phylogenetic studies

The theoretical isoelectric point and molecular weight of the protein were predicted by ExPASy software [55]. Predictions of a signal peptide for secretion and the cleavage site were performed with SignalP software [56]. The prediction of protein localization site was performed by PSORT II software [57]. NetNGlyc 1.0 software was used to predict N-linked glycosylation sites. Protein structure of Hp-FAR-2 (UniProt A0A183GLK6) was retrieved from AlphaFold Protein Structure Database [58]. Sequence similarity comparisons were conducted using BLASTP software [59]. The amino acid sequences of two FAR proteins from *Heligmosomoides polygyrus*, Hp-FAR-2 and Hp-FAR-1, were aligned using MUSCLE software [60] and visualized by Jalview [61]. Based on the amino acid sequences of Hp-FAR-2 and 27 other FAR proteins from 17 species of nematodes, a phylogenetic tree was constructed using maximum likelihood with LG + G model of amino acid substitution using Molecular Evolutionary Genetics Analysis (MEGA) software [62].

### Recombinant protein expression and purification

A codon-optimized synthetic gene encoding the mature Hp-FAR-2 protein, lacking the signal peptide (corresponding to amino acids 18–180 of the full-length gene), and fused to an N-Terminal 6-His-tag was cloned into expression vector pET histev N'term (provided by Dr. Martin Boulanger, University of Victoria, Victoria, CA) and transformed chemically into *Escherichia coli* BL21 (DE3) cells (Thermo Fisher Scientific, Catalog # EC0114). The recombinant *E. coli* cells were grown in ZYP-5052 auto-induction medium [63] with 50 µg/mL ampicillin (Sigma-Aldrich, Catalog # A5354) for 4 hours at 37°C. Expression

of Hp-FAR-2 was induced by auto induction at 30°C for 16 hours. Following induction, cells were harvested by centrifugation at 6,300 rpm for 15 minutes at 4°C. The cell pellet was stored in 50 mL conical tubes at −80°C until further purification. Once thawed, *E. coli* cells were lysed by French pressing twice. The lysate was centrifuged at 13,000 rpm for 10 minutes, at 4°C to remove cellular debris and isolate the soluble fraction of the lysate. Recombinant Hp-FAR-2 was purified from the soluble lysate by Ni-NTA bead batch binding for a total of three rounds. This was followed by size exclusion chromatography in a HiLoad 16/600 Superdex 75 prep grade column (GE Healthcare Life Sciences, Catalog # 28989333) equilibrated with size-exclusion chromatography (SEC) buffer (20 mM HEPES (2-[4-(2-hydroxyethyl)piperazin-1-yl]ethanesulfonic acid), 150 mM NaCl, pH 8.0). The purified protein was further analyzed on SDS-PAGE gel for verification of purity. Endotoxin level was quantified using the Pierce Chromogenic Endotoxin Quant Kit (Thermo Scientific, Catalog # A39552) and was below 0.01 EU/mL. The final purified protein was stored in SEC buffer (20 mM HEPES, 150 mM NaCl, pH 8.0) at −80°C.

### Fly husbandry and maintenance

All fly strains were cultivated on D2 glucose medium obtained from Archon Scientific in Durham, North Carolina. The flies were maintained at a temperature of 25°C with 50% humidity, following a 12-hour light and 12-hour dark cycle.

### Fly transgenesis

The Hp-FAR-2 amino acid sequence was codon-optimized for *Drosophila melanogaster* expression (IDT Codon Optimization Tool 2025). The *Drosophila* gene expression vector used to overexpress Hp-FAR-2 in our study, pUASTattB-5×UAS/mini_Hsp 70>{BglII-BmCyp15c1-XhoI} (VectorBuilder ID: VB220523–1438sdh), was constructed to include the Hp-FAR-2 transgene and packaged by VectorBuilder (Chicago, IL). Transgenic flies were generated by BestGene (Chino Hills, CA) via the PhiC31 site-specific serine integrase method by injecting the Hp-FAR-2 plasmid into the attP line y[1] w[67c23]; P{CaryP}attP2 (BDSC # 8622) and the attP line y[1] w[67c23]; P{CaryP}attP40 embryos. The transgenic UAS lines were then crossed with the TubP-GAL4 (BDSC #5138; strong, ubiquitous somatic expression) driver obtained from the Bloomington Drosophila Stock Center (BDSC) and screened. Transgenic UAS lines and the driver were crossed with w[1118] (BDSC # 3605) as the controls. Fly husbandry and crosses were performed under standard conditions at 25° C. Male and female adult flies aged 5–7 days were used for injections.

### Verification of transgene via mass spectrometry

Mass spectrometry was used to verify the expression of Hp-FAR-2 in transgenic fly lines. Transgenic UAS virgin female flies were crossed with male Tub-GAL4 flies, and the F1 generation was collected and screened for further downstream analysis. A total of 13 flies were ground in liquid nitrogen with a porcelain mortar and pestle. Next, 300 µL of 1X phosphate-buffered saline (PBS) was added to the fly tissue homogenate and aspirated by pipette into a 1.5 mL microcentrifuge tube. The sample was incubated with shaking at 4°C for 30 minutes then centrifuged for 10 minutes at 12,000 rpm. Fifty microliters of the supernatant, containing the soluble protein fraction, was aspirated into a new 1.5 mL microcentrifuge tube and used for further downstream analysis. Subsequently, 10 µL of 6X Gel loading dye was added to the soluble protein fraction. The sample was vortexed for 1 minute and placed on a heating block at 99°C for 5 minutes. This process was repeated two more times, for a total of three cycles. The sample was centrifuged at 6,000 rpm for 5 minutes, after which 15 µL of the supernatant was loaded onto a 15–20% gradient polyacrylamide gel prepared in-house. The gel was run in a Mini-PROTEAN Tetra Vertical Electrophoresis Cell (Bio-Rad, Catalog # 1658004) at 90 V for 110 minutes.

### Band excision and preparation of samples for mass spectrometry

The differential band from the gel lane representing the protein from the mutant fly (corresponding to the positive control protein band) was excised into small pieces (~1 mm) using a surgical blade and placed in a 1.5 mL low-binding Eppendorf tube for destaining with 10% methanol and 10% acetic acid. The gel pieces were then washed in three

alternating cycles of 500 µL of 50% acetonitrile in 50 mM ammonium bicarbonate (ABC) and 50 mM ABC, each for 10 minutes, followed by drying with centrifugal evaporation (SpeedVac). Protein reduction was performed by adding 200 µL of 20 mM DTT in 50 mM ABC, incubating at 37°C for 1 hour, followed by alkylation with 500 µL of 0.5 M iodo-acetamide in 50 mM ABC in the dark at room temperature for 30 minutes. The gel pieces were washed again in two cycles of 50% acetonitrile in 50 mM ABC, each for 10 minutes, then completely dried using centrifugal evaporation (~30 minutes). Protein digestion was carried out by adding mass spectrometry-grade trypsin (20 mM) in 200 µL of 50 mM ABC, incubating overnight (~16–19 hours) at 37°C in a water bath. The following day, the trypsin-digested sample was transferred to a new tube, while the original tubes containing gel pieces were refilled with 200 µL of 50% acetonitrile in 50 mM ABC, gently shaken on a thermomixer for 30 minutes, and then combined with the tryptic digested samples. Finally, the tryptic peptides were lyophilized in 0.1% formic acid before analysis using liquid chromatography with tandem mass spectrometry (LC-MS/MS).

## Mass spectrometry

LC-MS/MS was performed at the Proteomics Core Facility the University of California, Riverside, funded by NIH grant S10 OD010669. The sample was resuspended in 20 µL water containing 0.1% formic acid. Peptides were first separated by nano-LC (EASY-nLC 1200 system, Thermo Scientific) and analyzed by on-line electrospray tandem mass spectrometry (Orbitrap Fusion Tribrid Mass Spectrometry equipped with an EASY-Spray ion source). 5 µL peptide sample was loaded onto the trap column (Thermo Scientific Acclaim PepMap C18, 75 µm x 2 cm) with a flow of 10 µl/min for 3 minutes and subsequently separated on the analytical column (Acclaim PepMap C18, 75 µm x 25 cm) with a linear gradient, from 3% B to 37% B in 180 minutes. The column was re-equilibrated at initial conditions for 5 minutes. The flow rate was maintained at 300 nL/minute and column temperature was maintained at 45 °C. The electrospray voltage of 2.2 kV versus the inlet of the mass spectrometer was used. The Orbitrap Fusion Mass Spectrometry was operated in the data-dependent mode to switch automatically between MS and MS/MS acquisition. The raw data were analyzed using MaxQuant mass spectrometric data analysis software using reference protein sequence in addition to the contaminants as a database.

## Bacterial stock maintenance

Methods are adapted from [64]. *Streptococcus pneumoniae* (*S.p.*) was grown by shaking in glass vials with 5 mL tryptic soy (TS) broth (Difco Laboratories, MI, USA, Catalog # 211825), catalase (Worthington Biochemical Corporation, USA, Catalog # C9001052), and streptomycin (Fisher Scientific, Catalog # BP910–50) at 37°C with 5% $CO_2$ for 16–18 hours. The culture was diluted in catalase (100 µL) and TS broth to yield a final volume of 20 mL in a flask and incubated shaking until the OD600 ~ 0.4 (about 1 hour). The culture was then diluted again to a final volume of 50 mL, with 150 µL catalase, and incubated until the OD600 ~ 0.2–0.4. 5% glycerol was added to the final culture and stored in 1 mL aliquots at -80°C. To use the aliquots, one tube was thawed, spun down at 14,000 rpm for 4.5 minutes, the supernatant was removed, and the pellet was resuspended in the desired amount of PBS (160–180 µL yields ~ 100,000 CFUs) and serially diluted to yield the appropriate CFU doses. For quantification of CFUs, *S.p.* was plated on tryptic soy agar plates (Difco Laboratories, MI, USA, Catalog # 236950) supplemented with 50 mL/L sheep's blood (HemoStat Laboratories, Dixon, CA, USA, Catalog # 50-863-755) and 200 mg/L streptomycin at 37°C with 5% $CO_2$ for 16–18 hours. *Listeria monocytogenes* (*L.m.*) (serotype 4b, 19115, (ATCC, VA)) was also grown in batches in brain heart infusion (BHI) medium (Difco Laboratories, MI, USA, Catalog # 299070) at 37°C in aerobic conditions. Cultures were grown overnight in a flask inoculated with a fresh colony and diluted again under log phase (below OD600 ~ 0.2) and grown up to the desired OD600 (~0.4). The entire volume was transferred to a 50 mL conical tube for vortexing. Before freezing, a 5% glycerol solution was added to the culture and 1 mL aliquots were stored at -80°C. To use the aliquots, one tube was thawed, spun down at 14,000 rpm for 5 minutes, the supernatant was removed, and the pellet was resuspended in the desired amount of PBS (90–100 µL yields ~ 100,000

CFUs) and serially diluted to yield the appropriate CFU doses. For quantification of CFUs, *L.m.* was plated on BHI agar plates at 37°C for 16–18 hours.

## Fly injections, survival, and CFU quantification

Male Oregon-R flies aged 5–7 days were used for injections. The flies were anesthetized using $CO_2$ and injected with various colony-forming unit (CFU) doses of *S. pneumoniae*. The injections were performed with precise control, delivering a total volume of 50 nL, using a MINJ-FLY high-speed pneumatic injector (Tritech Research, CA) and individually pulled glass needles calibrated for accuracy. Injection sites were targeted close to the junction of the abdomen and thorax, slightly ventral from the dorsal-ventral cuticle axis, which could be easily observed beneath the haltere. Flies were injected with the CFU dose or a control of PBS and then placed in groups of 33 per vial. Flies injected with *S. pneumoniae* were maintained at a temperature of 28°C with 50% humidity. Daily monitoring of fly mortality was conducted for twenty days post-injection. For survival studies, one biological replicate is 60 flies and experiments were carried out with three biological replicates per treatment, totaling 180 flies. Kaplan-Meier survival curves were generated using GraphPad Prism software, and statistical analysis was performed using log-rank analysis (Mantel-Cox).

0-hour and 24-hour CFUs were determined by homogenizing a single fly in 200 μL PBS. Serial dilutions were performed and plated on the appropriate agar plates and incubated overnight. For CFU quantification, eight flies were used per replicate, and all experiments were triplicated per treatment, totaling 24 flies. Using GraphPad Prism software, results are shown as scatter plots with a line at the median. Statistical significance analyzed using a non-parametric unpaired t-test (Mann-Whitney).

## In vitro fluorescent based assays

The fatty acid and retinol binding preferences of Hp-FAR-2 were measured using the saturated fatty acid fluorescent probe 11-(Dansylamino) undecanoic acid (DAUDA) (Sigma-Aldrich, MO, USA Catalog # 39235) and retinol (Cayman Chemicals, USA Catalog # 20241). Fluorescent emission spectra for Hp-FAR-2 bound to DAUDA and retinol were measured at 25°C in a Corning 96-well black clear bottom plate (MilliporeSigma, MA,USA Catalog # CLS3631) yielding a total volume of 200 μL with an excitation of 345 nm and 350 nm respectively using a SpectraMax iD3 and iD5 Multi-Mode Microplate Reader. When DAUDA is encompassed by a binding protein and excited at 345 nm, a 50 nm blue shift in fluorescent emission is observed. The equilibrium dissociation constant (Kd) for Hp-FAR-2 bound to DAUDA was estimated by adding increasing concentrations of Hp-FAR-2 (0–20 μM), in 2 μM increments, to 1 μM DAUDA in PBS. When retinol is bound to a binding protein and excited at 350 nm, the fluorescence emission is greatly increased. The Kd for Hp-FAR-2 bound to retinol was estimated by adding increasing concentrations of retinol in 2 μM increments (up to 20 μM), to 1 μM Hp-FAR-2 in PBS. All ligand-binding experiments were conducted with three technical replicates. All data were normalized to the peak fluorescence intensity of DAUDA or retinol bound to FAR (yielding a value of 1) and plotted as relative fluorescence. Kd graphs were plotted as relative fluorescence and a nonlinear fit via the one site-specific binding equation was used to find the Kd value in GraphPad Prism. DAUDA was made from a stock solution of 10 mM in 100% ethanol then diluted in PBS to achieve the appropriate working stock of 10 μM. Retinol was freshly diluted to a working concentration and used within 24 hours of preparation of stock.

## In vitro fluorescent based competition assay

Competition studies were done by measuring the decrease in peak emission of DAUDA in the presence of another fatty acid in either 10-fold excess or equal concentration to DAUDA with an excitation of 345 nm. Linoleic acid, oleic acid, arachidonic acid, palmitic acid, eicosapentaenoic acid, docosahexaenoic acid, and α-linolenic acid were tested at concentrations of 10 μM or 1 μM and all lipids were obtained from Cayman Chemicals, MI, USA. All competition experiments

were replicated three times per treatment, plotted as bar graphs with individual points for each replicate. The data were normalized to the peak fluorescence intensity of DAUDA to FAR (yielding a value of 1) and corrected for background fluorescence of PBS alone. Data were plotted as bar graphs with error bars depicting mean with SEM. Statistics shown as ordinary one-way ANOVA with Dunnett's multiple comparisons test done in GraphPad Prism. All fatty acids (except α-linolenic acid) were stored in -20°C and freshly diluted before each experiment in PBS to a working concentration. α-linolenic acid was used directly from the stock.

## Phenoloxidase activity

Flies were injected with 1,000 CFUs of *L. monocytogenes* to elicit an immune induced melanization cascade. Phenoloxidase (PO) activity was measured as previously described [64]. To collect hemolymph, 50 flies 6 hours post-injection were pricked through the thorax and placed in a pierced 0.5 µL Eppendorf tube. Samples were centrifuged at 10,000 g for 20 minutes at 4°C. Using a clear 96-well plate, each well contained 160 µL of 3 mg/mL 3,4-Dihydroxy-L-phenylalanine (L-DOPA) (Sigma-Aldrich, Catalog # D9628) dissolved in a buffer solution of 37.5% potassium phosphate (1 M) and 62.5% sodium phosphate (1 M), pH 6.5, 4 µL of hemolymph sample and 5 µL $CaCl_2$ (20 mM). PO activity was measured by kinetic reads at 29°C at 492 nm every minute for 180 minutes with 3 seconds of shaking between reads. Data were taken at the peak OD value (timepoint ~150 minutes) with the OD of L-DOPA alone subtracted from all biological values. One biological replicate is 50 flies and experiments were carried out with three biological replicates per treatment, totaling 150 flies. Data were plotted as bar graphs with error bars depicting mean with SEM. Statistics shown as ordinary one-way ANOVA with Dunnett's multiple comparisons test done in GraphPad Prism.

## pHrodo Phagocytosis

Methods are adapted from [64]. Phagocytic activity was measured using the pHrodo Red *E. coli* BioParticles Conjugate for Phagocytosis (Thermo Fisher Scientific, Catalog # P35361) and pHrodo Red *S. aureus* BioParticles Conjugate for Phagocytosis (Thermo Fisher Scientific, Catalog # A10010). The pHrodo Red *E. coli* and pHrodo red *S. aureus* were each resuspended in PBS to a working concentration of 4 mg/mL then diluted 1:4 in PBS for injection. Flies were injected with pHrodo Red *E. coli* (1 mg/mL) in PBS or pHrodo Red *E. coli* (1 mg/mL) plus 150 ng Hp-FAR-2. The same procedure was done for pHrodo Red *S. aureus.* Injected flies were incubated at 28°C for 1 hour. To prepare for imaging, flies were placed in -80 °C for 10 minutes then fly wings were removed using micro dissecting scissors and placed on ice for subsequent imaging. The dorsal side of the abdomen was imaged with an X-Cite 120Q fluorescence lamp, and a ZEISS Axiocom 506 Color microscope camera attached to a ZEISS SteREO Discovery V12 microscope at 10x magnification. ImageJ software [65] was used to measure area-normalized total fluorescence of isolated red channels. Prior to analysis, images were pre-processed and denoised (median filter, 2-pixel radius) and background-subtracted (rolling ball subtraction, 50-pixel radius). Thresholding was used for segmentation to refine the objects using intensity-threshold algorithm in FIJI (Shanbhag, 30, 255). After processing, the areas containing fluorescent signals from phagocytosis were measured and area-normalized and reported as relative fluorescence. A macro comprising the sequence of ImageJ commands used for image analysis is provided in the supplementary information (S1 Data). Experiments were biologically triplicated with at least three flies per biological replicate. Data shown as bar graphs with individual points representing all experimented flies. Statistical significance was analyzed using an unpaired t-test with error bars depicting mean with SEM done in GraphPad Prism.

## Antimicrobial peptide gene expression – Quantitative PCR

Total RNA of 15 flies per treatment were extracted 18 hours post-injection using TRIzol reagent (Invitrogen, Carlsbad, California, USA) according to manufacturer's instructions. Prior to cDNA synthesis, RNA was treated with DNase I (New

England Biolabs, Carlsbad, CA, USA Catalog # M0303) to remove traces of gDNA according to the manufacturer's instructions. Concentration and purity of RNA was determined by nanodrop (Thermo Scientific NanoDrop 2000c, Catalog # ND-2000C). RNA was reverse-transcribed into first-strand cDNA and primed with oligo(dT) using the LunaScript Reverse Transcriptase SuperMix (New England Biolabs, Carlsbad, CA, USA Catalog # NEB-E3010) following the manufacturer's instructions, in a MultiGene OptiMax Thermal Cycler (Labnet international, NJ). Quantitative PCR (qPCR) was performed using the PerfeCTa SYBR Green FastMix (VWR—Quanta Bio Sciences, MD, USA Catalog # 95072–250) following the manufacturers protocol in a final volume of 10 µL for each qPCR reaction. Gene specific primers for *Defensin, Drosomycin, Diptericin, Metchnikowin* and the housekeeping gene *Tubulin* (Integrated DNA Technologies, IA) were added to a final concentration of 500 nM. All the samples were run in triplicate in a thermocycler (CFX96 Touch Real-Time PCR Detection Systems, Hercules, California, USA) with the following steps: 95 °C for 30 seconds, followed by 40 cycles of 95 °C for 15 seconds, 55 °C for 30 seconds and 68 °C for 30 seconds. Relative normalized expression was calculated by the ΔΔCq method using the housekeeping gene. Experiments were carried out with three biological replicates per treatment, totaling 45 flies. Plots shown as bar graphs, with error bars depicting mean with SEM. Statistics shown as a two-way ANOVA with Dunnett's multiple comparisons test done in GraphPad Prism. All primer sequences are listed in the supplemental information (S2 Table).

## RAW 264.7 Macrophage assays

**Cell culture maintenance and passaging.** The RAW 264.7 mouse macrophage cell line was purchased from the American Type Culture Collection (ATCC, TIB-71). RAW 264.7 cells were cultured in Dulbecco's Modification of Eagle's Medium with L-Glutamine 4.5g/L Glucose and Sodium Pyruvate (Thermo Fisher Scientific, Catalog # 10–013-CV) supplemented with 10% Fetal Bovine Serum (Gibco, Catalog # A5670701), 100 U/mL Penicillin/Streptomycin (Gibco, Catalog # 15140148) and 0.15% Sodium Bicarbonate (Thermo Fisher Scientific Catalog # 25080094). The cell medium was filtered by a 0.22 µm pore-size filtration system (Corning Catalog # 431098) and stored at 4°C. Cells were grown in a 75 cm$^2$ flask (Sarstedt, Catalog # 83.3911.002) at 37°C and 5% $CO_2$. Cells were subcultured at 70% confluence by first decanting the supernatant and washing the cells with sterile PBS (Sigma-Aldrich Catalog # D537) twice. Fresh media is added, and cells are harvested by gently scraping the flask once using a cell scraper. Cell suspension was transferred to a 15 mL sterile conical tube and centrifuged at 300 g for 7 minutes. Cells were then subcultured at a 1:6 ratio in fresh cell medium and incubated again at 37°C and 5% $CO_2$. Cell culture medium was replaced every 2–3 days. All experiments were performed before the 10$^{th}$ passage.

## Macrophage polarization

Cells were harvested at 70–80% confluence for polarization by first decanting the supernatant and washing the cells with sterile PBS twice. Cells were detached by gentle scraping, and the cell suspension was transferred to a 15 mL sterile tube and centrifuged at 300 g for 7 minutes. After centrifugation, the supernatant was removed, and cells were washed once more by adding 10 mL cell medium and resuspending the pellet. Cells were centrifuged at 300 g for 7 minutes, decanted, and resuspended in 6 mL complete medium. Cells were quantified using 0.4% Trypan Blue (Bio-Rad, Catalog # 1450022) by adding 10 µL cell suspension to 10 µL 0.4% Trypan Blue and loading 10 µL to a dual-chamber slide (Bio-Rad, Catalog # 1450019) where percent of live cells were quantified by a Bio-Rad TC20 automated cell counter (Bio-Rad, Catalog # 1450102). Cells were adjusted to a density of 6.25 x 10$^4$ cells/mL with cell medium and a 400 µL cell suspension was distributed to each well in a 48-well plate and incubated overnight at 37°C and 5% $CO_2$.

To generate M1 and M2 macrophages, cells were exposed to 400 µL cell medium containing 20 ng/mL IFN-γ and 100 ng/mL LPS or 40 ng/mL IL-4 respectively for 48 hours at 37°C and 5% $CO_2$. For M1 macrophages, LPS was added at the 47$^{th}$ hour. To assess whether Hp-FAR-2 influences macrophage polarization, cells were treated with corresponding

activating factor and either 3 µg/mL Hp-FAR-2 or PBS. After 24 hours, the supernatant was removed from each well and cells were resuspended in 350 µL RNeasy Lysis Buffer (Qiagen, Catalog # 79216) with β-mercapthenol. The lysates were transferred to a sterile 1.5 mL microcentrifuge tubes, flash frozen using liquid nitrogen and stored at -80°C until RNA extraction.

**Macrophage polarization expression– Quantitative PCR**

Total RNA was isolated from RAW 264.7 macrophages using the RNeasy Mini Kit (Qiagen, Catalog # 74104) according to manufacturer's instructions. Prior to cDNA synthesis, RNA was treated with DNase I (New England Biolabs, Carlsbad, CA, USA Catalog # M0303S) to remove traces of gDNA according to the manufacturer's instructions. RNA concentration and quality was determined by NanoDrop. RNA was reverse-transcribed into first-strand cDNA and primed with oligo(dT) using the LunaScript Reverse Transcriptase SuperMix (New England Biolabs, Carlsbad, CA, USA Catalog # E3010) following the manufacturer's instructions, in a MultiGene OptiMax Thermal Cycler (Labnet international, NJ). Quantitative PCR was performed using the PerfeCTa SYBR Green FastMix (VWR—Quanta Bio Sciences, MD, USA Catalog # 95072–250) following the manufacturers' protocol in a final volume of 10 µL for each qPCR reaction. Gene specific primers for *TNF-α, IL-6, Chil3, Arg1*, and the housekeeping gene glyceraldehyde-3-phosphate dehydrogenase (*GADPH*) (Integrated DNA Technologies, IA) were used, and all primers were added to a final concentration of 500nM. All the samples were run in triplicate in a thermocycler (CFX96 Touch Real-Time PCR Detection Systems, Hercules, California, USA) with the following steps: 95 °C for 30 seconds, followed by 40 cycles of 95 °C for 15 seconds, 55 °C for 30 seconds and 68 °C for 30 seconds. Relative normalized expression was calculated by the ΔΔCq method using the housekeeping gene. The experiment was carried out with three biological replicates per treatment with plots shown as bar graphs and error bars depicting mean with SEM. Statistics shown as a two-way ANOVA with Dunnett's multiple comparisons test done in GraphPad Prism. All primer sequences are listed in the supplemental information (Table S2).

**RAW 264.7 phagocytic activity**

The phagocytic activity of RAW 264.7 macrophages were determined using the pHrodo Red *E. coli* BioParticles Conjugate for Phagocytosis according to manufacturer's protocols. The pHrodo Red *E. coli* was resuspended in cell culture grade PBS to a concentration of 1 mg/mL. Briefly, $5.0 \times 10^4$ cells were seeded in a 24-well cell culture plate (Thermo Fisher Scientific, Catalog # 150628) with 500 µL complete medium containing 1 µg/mL Hp-FAR-2. Cells were incubated overnight at 37°C and 5% $CO_2$. After overnight incubation, pHrodo Red *E. coli* BioParticles was added to the cell medium at a final concentration of 0.1 mg/mL. Cells were incubated for 1.5 hours at 37°C. After incubation, the cells were fixed with 4% paraformaldehyde, and the nucleus was stained with 1 µg/mL DAPI (Thermo Fisher Scientific, Catalog # D1306). Fluorescent images of cells were acquired using an apotome microscope, with one replicate totaling ~1250 cells and four replicates measured per treatment. Phagocytosis efficiency was evaluated by measuring the mean fluorescence intensity using ImageJ software. Data is shown as bar graphs and error bars depicting mean with SEM. Statistics shown as an unpaired t-test done in GraphPad Prism.

**Statistics**

All statistics were done with GraphPad Prism 9.1.0 for Mac. Statistical significance indicated with asterisks indicating the following p-value cut offs: *$p < 0.05$, **$p < 0.01$, ***$p < 0.001$, ****$p < 0.0001$.*

## Results and discussion

### Identification, cloning, and expression of Hp-FAR-2

Previously uncharacterized peptide fragments matching a fatty acid and retinol binding protein (NCBI accession: VDP39696.1) were identified in the adult ES products of *H. polygyrus* in two independent studies [51,53] and in the ES

products of L4 *H. polygyrus* female and male mixed cultures [54]. As such, this FAR was of particular interest and is designated herein as Hp-FAR-2. Analysis of the Hp-FAR-2 gene sequence identified a 543 bp fragment encompassing an open reading frame (ORF), encoding a predicted protein of 180 amino acids. The cDNA corresponding to the mature Hp-FAR-2 was cloned into the expression vector pET histev N' term. Recombinant Hp-FAR-2 was purified from *E. coli* lysates using nickel resin affinity chromatography. Hp-FAR-2 was further purified using size-exclusion chromatography and SDS-PAGE showed that the recombinant Hp-FAR-2 was purified to homogeneity, with a single band present of approximately 17–18 kDa (S1 Fig).

## Sequence analysis of the Hp-FAR-2 protein

The mature Hp-FAR-2 protein encodes 163 amino acids with a theoretical mass of 18 kDa and theoretical isoelectric point of 8.6. Like other FARs, Hp-FAR-2 contains a secretory signal peptide and a casein kinase II phosphorylation site (S2 Fig) [38]. No N-linked glycosylation sites were predicted in the amino acid sequence of Hp-FAR-2 using NetNGlyc 1.0. Computer-based secondary structure analysis predicts a rich α-helix and coiled coil structure, similar to other nematode FARs [11,35,39]. Hp-FAR-2 was predicted to be in the extracellular compartment (including cell wall) [57]. Hp-FAR-2 sequence had the highest similarity with a FAR from rat hookworm, *Nippostrongylus brasiliensis* (NCBI accession: WKY01693.1, (91% over 180 residues, E-value 1e-110) and a FAR from the human hookworm, *Necator americanus* (NCBI accession: XP_013293703.1, 88% over 180 residues, E-value 6e-106).

Analysis of the *H. polygyrus* genome assembly projects (NCBI accession: PRJEB15396, PRJEB1203) using the WormBase ParaSite BioMart data-mining tool, filtering for the Gp-FAR-1 domain (Pfam ID: 05823), revealed six FAR- encoding genes (Table S1) [66]. Among these, the previously characterized Hp-FAR-1 (NCBI accession: AAK57805.1), and Hp-FAR-2 were included. The relationships of these proteins were evaluated using amino acid sequences of Hp-FAR-1, Hp-FAR-2, and 26 FAR proteins from 16 species of nematodes (Fig 1A). Hp-FAR-2 clustered with a FAR from *N. brasiliensis*, whereas Hp-FAR-1 appeared more isolated, indicating evolutionary divergence between the two, consistent with their placement in separate clades (Fig 1A). Furthermore, the amino acid sequences of Hp-FAR-1 and Hp-FAR-2 were aligned using MUSCLE and identical residues are shaded in orange (Fig 1B). Hp-FAR-2 shares 35% identity over 159 residues, illustrating that although these genes share the FAR protein domain, they diverge significantly in sequence.

To further understand the role of these FARs in infection, we analyzed data from previous studies and found that Hp-FAR-1 was identified as the third-most abundant FAR in the adult HES [51] and the most abundant FAR in L4 larvae [53,54], across three separate studies. In contrast, Hp-FAR-2 was identified as the most abundant FAR in the adult HES [51,53] and the second-most abundant FAR in L4 larvae [53,54]. We hypothesize Hp-FAR-1 and Hp-FAR-2 may play distinct roles in host parasitism due to differing abundance in the HES at various life stages, with Hp-FAR-1 being most abundant in the L4 stage and Hp-FAR-2 in adults.

## Fluorescent binding reveals Hp-FAR-2 affinity for fatty acids and retinol

To determine the ligand preferences of Hp-FAR-2 we used the saturated and fluorescent probe 1-(Dansylamino) undecanoic acid (DAUDA) and retinol in fluorescent-based ligand-binding assays. DAUDA is poorly fluorescent in PBS, but highly fluorescent when removed from a polar solvent in a protein binding site. Hp-FAR-2 was found to bind to DAUDA, indicated by the degree of blue shift in fluorescent emission between 1 µM DAUDA alone, shown in purple, and DAUDA with the addition of 14 µM Hp-FAR-2, shown in pink (from 544 nm to 486 nm respectively) (Fig 2A). This blue shift reflects the relocation of DAUDA from a polar aqueous environment into a highly apolar binding site within Hp-FAR-2, consistent with other FAR proteins [9,11,15,16,39]. The equilibrium dissociation constant (Kd) for Hp-FAR-2:DAUDA was determined by increasing the concentration of Hp-FAR-2 (0–20 µM) in the presence of 1 µM DAUDA (Fig 2B). We found the Kd for DAUDA and Hp-FAR-2 to be 3.16 µM, consistent within the range of dissociation constants (0.1 -10 µM) reported for transporter/shuttle proteins carrying fatty acids (Fig 2B) [67,68]. Hp-FAR-2 was found to bind to retinol, indicated by a

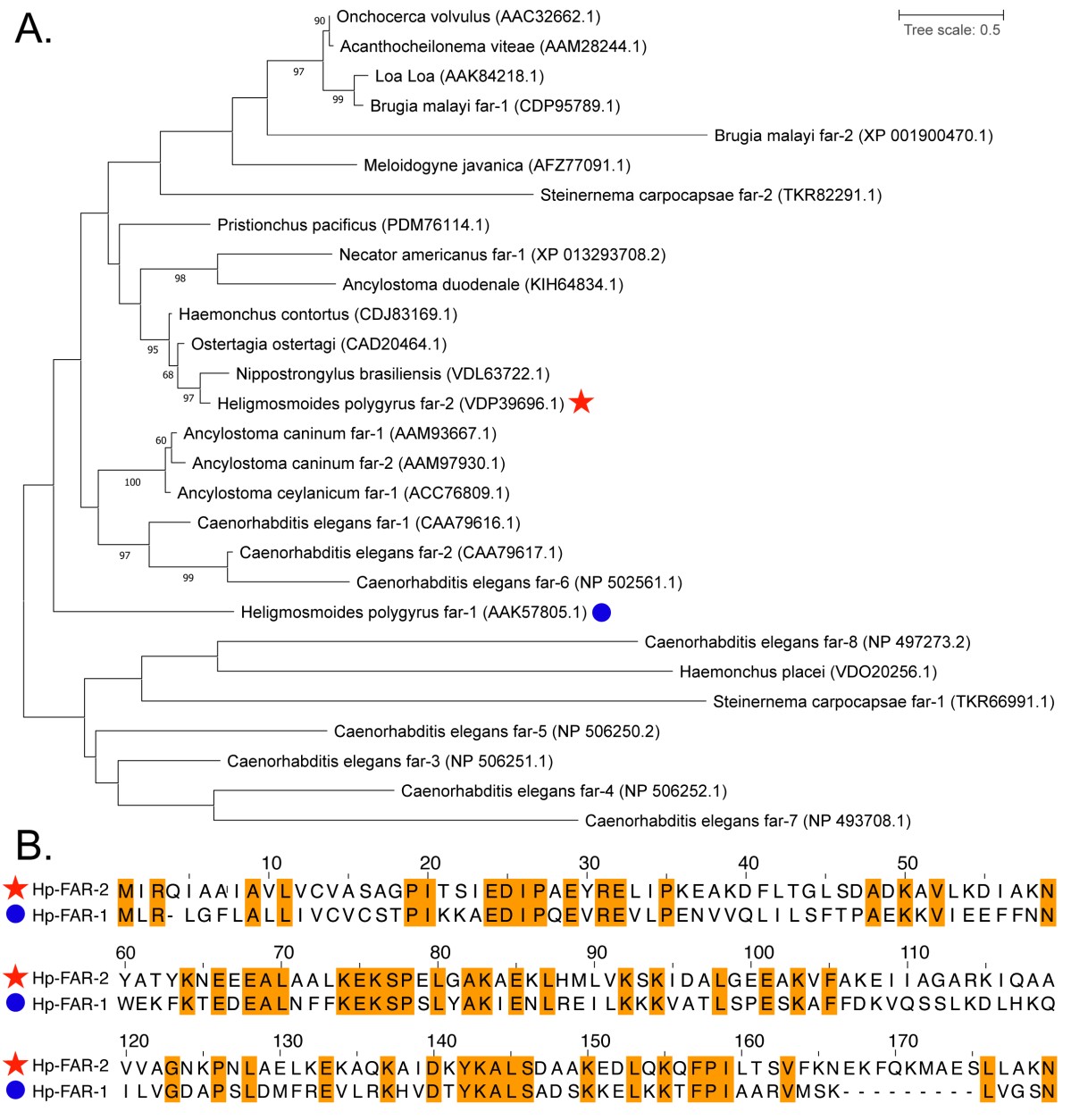

**Fig 1. Phylogenetic tree of fatty acid- and retinol-binding protein (FAR) of *Heligmosomoides polygyrus* with other nematode FARs. A)** The phylogenetic tree was constructed using amino acids sequences of FARs from *Nippostronglyus brasiliensis* (VDL63722.1), *Heligmosomoides polygyrus* (AAK57805.1, VDP39696.1), *Ostertagia ostertagi* (CAD20464.1), *Haemonchus contortus* (CDJ83169.1), *Necator americanus* (XP_013293708.2), *Ancylostoma duodenale* (KIH64834.1), *Pristionchus pacificus* (PDM76114.1), *Ancylostoma caninum* (AAM93667.1, AAM97930.1), *Ancylostoma ceylanicum* (ACC76809.1), *Caenorhabditis elegans* (CAA79616.1, CAA79617.1, NP_506251.1, NP_506252.1, NP_506250.2, NP_502561.1, NP_493708.1, NP_497273.2,), *Meloidogyne javanica* (AFZ77091.1), *Onchocerca volvulus* (AAC32662.1), *Acanthocheilonema viteae* (AAM28244.1), *Loa-Loa* (AAK84218), *Brugia malayi* (CDP95789.1, XP_001900470.1), *Steinernema carpocapsae* (TKR66991.1, TKR82291.1), and *Haemonchus placei* (VDO20256.1) obtained from NCBI protein database. Protein sequence alignment was done by MUSCLE. The LG + G model of amino acid substitution was used to construct a maximum-likelihood phylogenetic tree using MEGA version 11. Hp-FAR-1 is highlighted by a blue circle and Hp-FAR-2 is highlighted by a red star. Values on branches represent bootstrap values, and only values with bootstrap values of >60 are displayed. **B)** Protein sequence alignment of Hp-FAR-2 and Hp-FAR-1 by MUSCLE using MEGA version 11 and visualized by Jalview. Identical residues are shaded in orange.

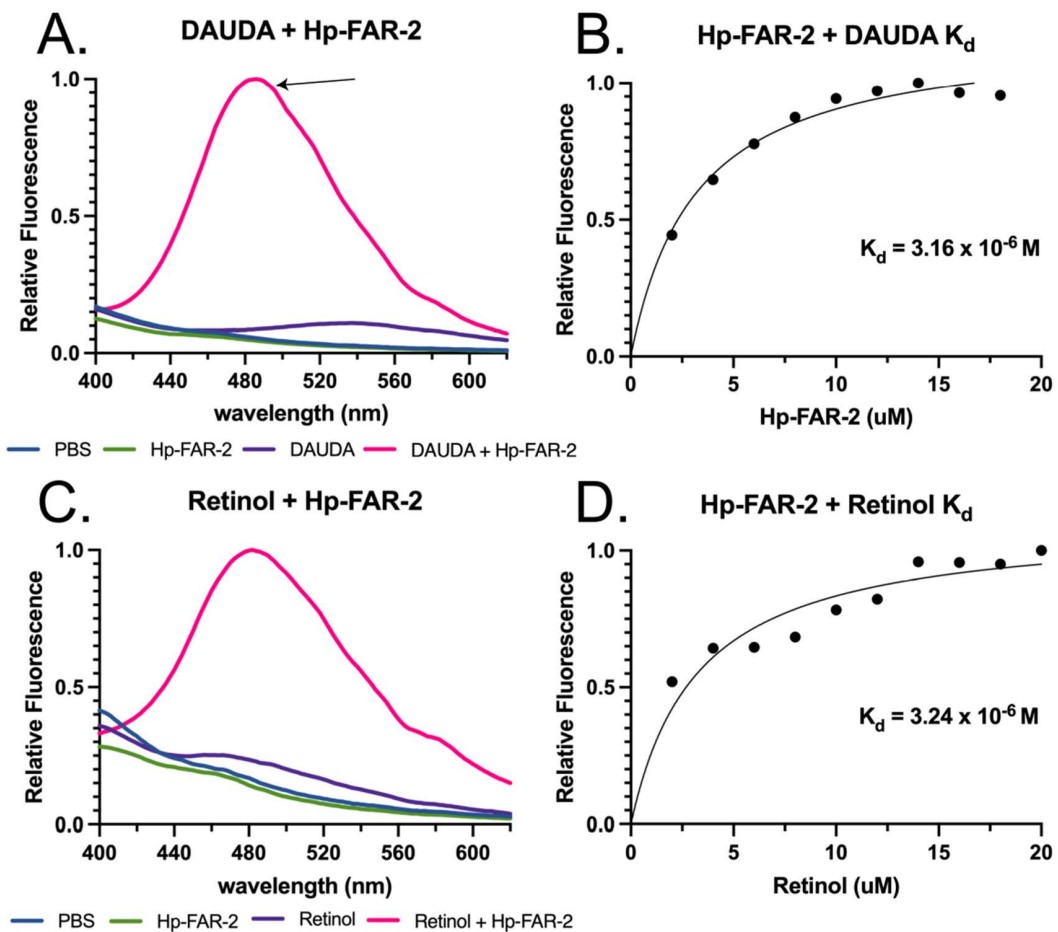

**Fig 2. In vitro binding kinetics of Hp-FAR-2.** Fatty acid and retinol binding affinity of recombinant Hp-FAR-2 were tested using saturated and fluorescent probe 1-(Dansylamino) undecanoic acid (DAUDA) and retinol in fluorescent-based ligand-binding assay. **A)** Fluorescence emission spectra for 1 μM DAUDA in buffer alone (purple) or DAUDA in buffer plus 14 μM Hp-FAR-2 (pink), when excited at 345 nm. When DAUDA is bound to FAR in vitro, a ~50 nm blue shift of fluorescent emission is observed in the peak as highlighted by the arrow. **B)** Titration curve for determining the dissociation constant (Kd) for interaction of DAUDA with Hp-FAR-2 reveal a Kd of 3.16 μM. **C)** Fluorescence emission spectra for 20 μM retinol in buffer alone (purple) or retinol with buffer plus 1 μM Hp-FAR-2 (pink), when excited at 350 nm. When retinol is bound to FAR in vitro, the peak fluorescence is greatly increased. **D)** Titration curve of increasing concentration of retinol in buffer binding to 1 μM Hp-FAR-2 reveal a Kd of 3.24 μM. All ligand-binding experiments were conducted with three technical replicates. All data were normalized to the peak fluorescence intensity of DAUDA or retinol bound to FAR (yielding a value of 1) and plotted as relative fluorescence. Kd was estimated using one site-specific binding in Graphpad Prism.

change in the fluorescence emission between 20 μM retinol alone (purple) and retinol with the addition of 1 μM Hp-FAR-2 (pink) (Fig2C). The Kd for Hp-FAR-2:retinol was determined by increasing the concentration of retinol (0–20 μM) in the presence of 1 μM Hp-FAR-2 (Fig 2D). We found the Kd for retinol and Hp-FAR-2 to be 3.24 μM (Fig 2D).

### Competitive binding assays identify Hp-FAR-2 exhibits binding affinity for omega-3 and omega-6 polyunsaturated fatty acids

Competition based assays were used to determine fatty acid binding preference of Hp-FAR-2 by measuring the degree of displacement of DAUDA, as indicated by a reduction in the fluorescence intensity at 486 nm in the presence of various saturated and unsaturated fatty acids. Firstly, we measured the degree of displacement of DAUDA in the presence

of a tenfold excess concentration of DAUDA to fatty acid (1:10). Various saturated and unsaturated fatty acids with chain lengths 16 and 18–22 were tested including arachidonic acid (AA), oleic acid (OA), linoleic acid (LA), palmitic acid (PA), eicosapentaenoic acid (EPA), docosahexaenoic acid (DHA) and α-linolenic acid (α-L). In tenfold excess, all tested fatty acids displaced DAUDA except PUFA C18, α-L, with maximal DAUDA displacement in the presence of monounsaturated fatty acid C18, OA (Fig 3A). We then used an equal competitive ratio of DAUDA to fatty acid (1:1), which provided a more sensitive indication of fatty acid preference. In equal concentration of DAUDA to fatty acid (1:1), Hp-FAR-2 binds to PUFAs with chain lengths 18 and 20, with maximal DAUDA displacement in the presence of OA, followed by PUFA C:18 LA, and finally PUFA C:22 DHA (Fig 3B). The results suggest that Hp-FAR-2 has various fatty acid binding preferences and is not solely dependent on chain length.

While many studies on FARs have tested fatty acid preferences at approximately ten-fold excess of fatty acid relative to DAUDA [16,18,19], testing at a more competitive ratio indicated that Hp-FAR-2 has binding preferences for both omega-3 and omega-6 PUFAs, specifically those which are precursors to bioactive lipid mediators called oxylipins in mammals [20]. PUFAs and their downstream lipid mediators can affect the onset, development, and resolution of inflammation. Cell culture studies demonstrate that omega-3 PUFAs EPA and DHA can inhibit the production of inflammatory cytokines like TNF-α, IL-1β, IL-6 and IL-8 by monocytes, macrophages, and endothelial cells [69–71]. In contrast, pro-inflammatory cytokines stimulate the release of omega-6 PUFA AA, which is then converted into prostaglandins that contribute to inflammation [72]. Oleic acid is a monounsaturated fatty acid that has also been reported as an anti-inflammatory fatty acid involved in the activation of different pathways of immune competent cells [73]. Oleic acid has been shown to enhance Treg-suppressive function [74], reduce pro-inflammatory cytokines [75], and reduce the migration of neutrophils

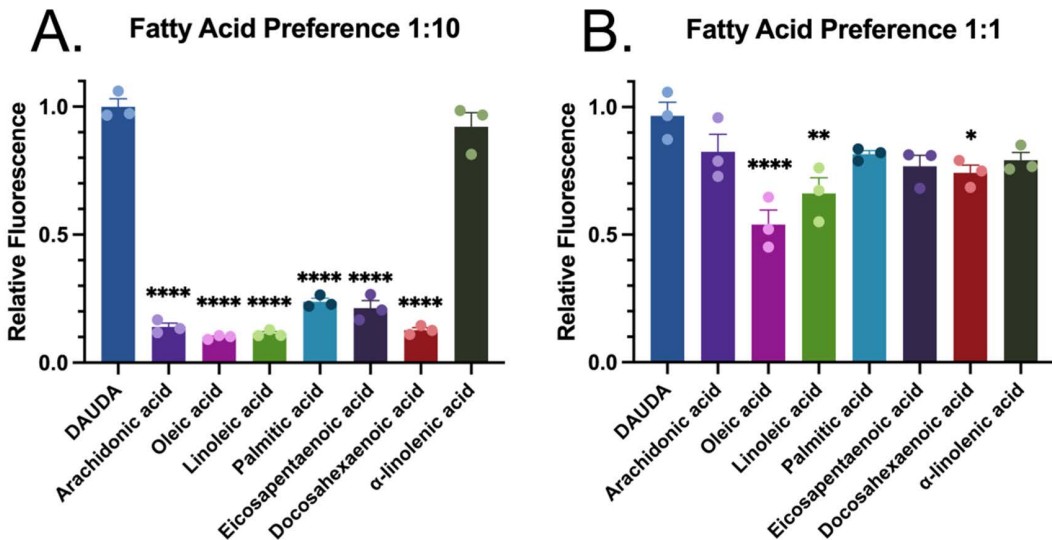

**Fig 3. In vitro competition assays with various fatty acids reveal a binding preference for both omega-3 and omega-6 polyunsaturated fatty acids.** Fatty acid binding preference of recombinant Hp-FAR-2 were tested using saturated and fluorescent probe 1-(Dansylamino) undecanoic acid (DAUDA) and in fluorescent-based ligand-binding assay. **A)** Competitive binding assays with 10-fold excess of fatty acid relative to DAUDA (1:10) were performed by adding 10 μM of various fatty acids—arachidonic acid, oleic acid, linoleic acid, palmitic acid, eicosapentaenoic acid, docosahexaenoic acid, or α-linolenic acid to 1 μM DAUDA and Hp-FAR-2. All fatty acids except α-linolenic acid significantly displaced DAUDA compared to DAUDA alone (dark blue). **B)** Competitive binding assays with equal concentration of DAUDA to fatty acid (1:1) were performed by adding 1 μM of various fatty acids to 1 μM DAUDA and Hp-FAR-2. Oleic acid, linoleic acid, and docosahexaenoic acid all significantly displaced DAUDA as seen by a reduction in relative fluorescence compared to DAUDA alone indicating a fatty acid preference. All competition experiments were replicated 3 times per treatment, plotted as bar graphs with individual points for each replicate. Statistics were analyzed using a one-way ANOVA, with error bars depicting the mean with SEM (standard error of the mean). Asterisks indicate the following p-value cut offs: * p<.05, ** p<.01, **** p<0.0001.

[76]. Furthermore, omega-3 and omega-6 fatty acids have been shown to mediate the growth and survival of protozoan parasites through anti-inflammatory mediators such as resolvins and lipoxins, which have demonstrated both in vivo and in vitro protective effects against various protozoan infections [77]. PUFAs have been shown to modulate the host immune response by promoting a shift towards a more effective immune defense against parasitic invaders through the regulation of inflammatory mediators such as prostaglandins, leukotrienes, and thromboxanes, which play a role in controlling the inflammatory reaction [77]. We hypothesize that Hp-FAR-2 may be released during infection to sequester omega-3 and omega-6 fatty acids, aiding in parasite growth and survival and modulating inflammatory reactions in the host. Furthermore, FARs have been hypothesized to function in nutrient acquisition, as parasitic nematodes are unable to synthesize fatty acids and retinol de novo; however, this role remains unconfirmed since no evidence has shown that FARs are ingested or reabsorbed by parasites.

## Hp-FAR-2 does not affect survival against bacterial infection in *Drosophila melanogaster*

We investigated the immunomodulatory effects of Hp-FAR-2 using a *Drosophila* infection model. *Drosophila* is a powerful model to study innate immunity in eukaryotic organisms, as it shares conserved innate immune pathways with mammals [78]. It allows for larger sample sizes and greater availability of transgenic lines compared to mammalian models. Additionally, PUFAs have been found to modulate *Drosophila* immunity and they can utilize immunostimulatory lipid signaling to mitigate bacterial infections. Specifically, LA was found to stimulate phagocytosis by hemocytes, while both LA and AA increased antimicrobial peptide expression when *D. melanogaster* is exposed to a heat-killed bacterial pathogen [64]. This model was successfully utilized to elucidate the function of two FARs from the entomopathogenic nematode *S. carpocapsae* in host immunity as well as FARs from *C. elegans* and *A. ceylanicum*, all of which were shown to be potent modulators of host immunity [18]. First, we evaluated if injection of recombinant Hp-FAR-2 was toxic in *D. melanogaster* by injecting 5–7-day adult male Oregon-R flies with either 200 ng of Hp-FAR-2 or PBS and observed survival for twenty days. No significant change in survival was observed in Hp-FAR-2 injected flies compared to the PBS-injected control group, indicating Hp-FAR-2 is not toxic to flies (S3 Fig). To determine immune modulation by Hp-FAR-2, we investigated if Hp-FAR-2 would influence *D. melanogaster* survival against bacterial infection of Gram-positive extracellular pathogen *Streptococcus pneumoniae* (*S.p.*). At a high dose, *S. pneumoniae* can kill flies rapidly, facilitating quick assessment of bacterial load and host survival [79]. We established that injection of male flies with an $LD_{40}$ (dose that kills 40% of flies within 2–5 days post-infection) of *S. pneumoniae* provided sufficient sensitivity to detect shifts in infection outcome, measured as survival and microbial load at 0 hour and 24 hours post-injection. To investigate the immunomodulatory effects of Hp-FAR-2, male flies were co-injected with either 200 ng Hp-FAR-2 plus an $LD_{40}$ of *S. pneumoniae* or an $LD_{40}$ of *S. pneumoniae* alone. No significant change in survival was observed in the $LD_{40}$ of *S. pneumoniae* plus Hp-FAR-2 co-injected flies compared to *S. pneumoniae* alone (Fig 4A). Furthermore, no significant change in microbial growth 24 hours post-injection was found between the two groups, collectively indicating Hp-FAR-2 does not affect fly survival against a $LD_{40}$ of *S. pneumoniae* (Fig 4B). We further investigated whether Hp-FAR-2 influenced survival at a higher dose of *S. pneumoniae* by challenging male flies with a $LD_{60}$ (dose that kills 60% of flies within 2–5 days post-injection) of *S. pneumoniae* or a $LD_{60}$ plus 200 ng Hp-FAR-2 and recording microbial load 24 hours post- injection and survival. No significant change in survival was observed in $LD_{60}$ of *S. pneumoniae* plus Hp-FAR-2 co-injected flies compared to $LD_{60}$ of *S. pneumoniae flies* alone (Fig 4C). Furthermore, no significant change in microbial growth 24 hours post-injection was found between the two groups, indicating Hp-FAR-2 also does not affect fly survival against a $LD_{60}$ of *S. pneumoniae* (Fig 4D).

In addition to co-injection experiments, we used two transgenic fly lines expressing Hp-FAR-2 via the GAL4/UAS system to further assess the impact of Hp-FAR-2 on infection outcome. Fly lines were generated with the UAS-Hp-FAR-2 transgene inserted into either chromosome two (Ch2-UAS) or chromosome three (Ch3-UAS) to control for positional effects. Ectopic expression of Hp-FAR-2 was induced using the ubiquitous Tubp-GAL4 driver and crosses were termed TubP > Ch2-UAS and TubP > Ch3-UAS. Expression was confirmed using mass spectrometry (Table S3). Driver and UAS

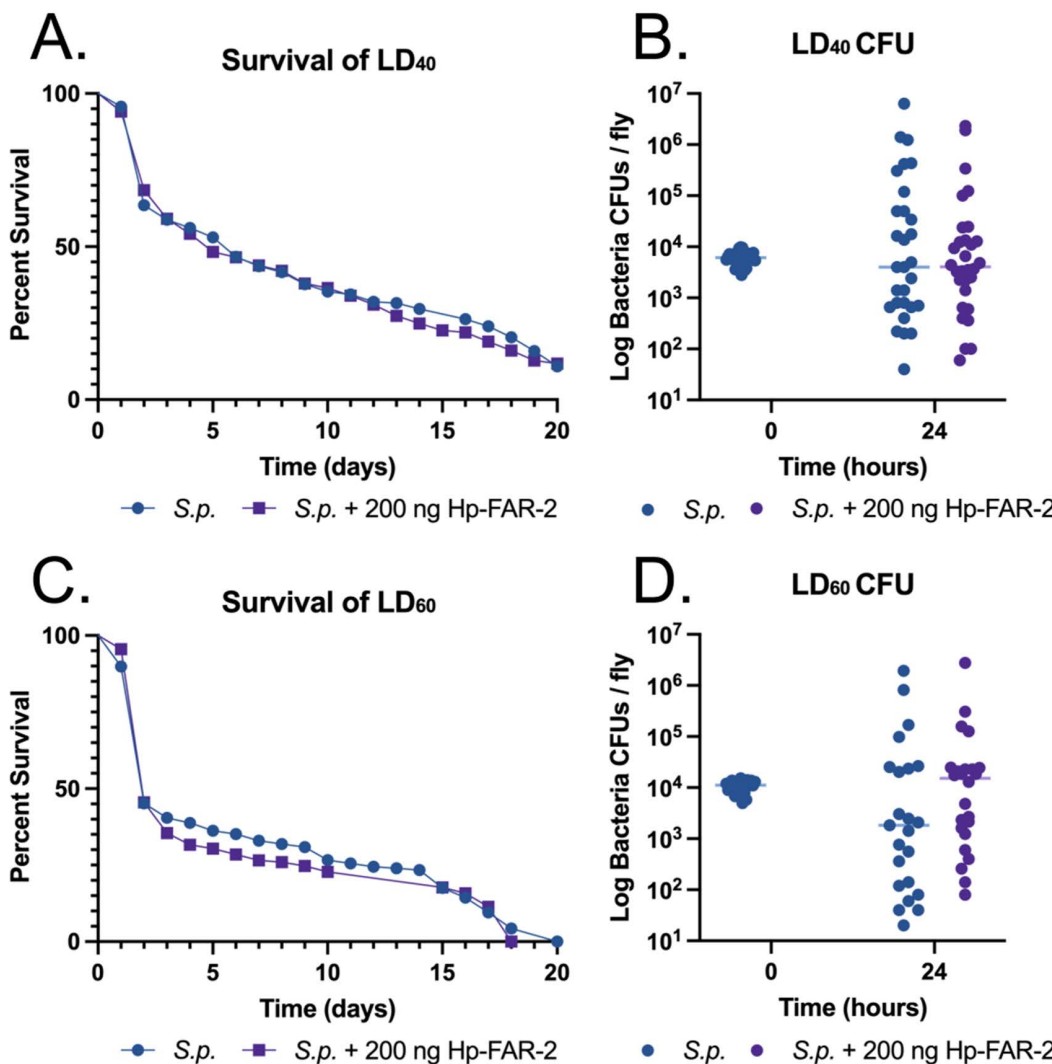

**Fig 4. Recombinant Hp-FAR-2 does not influence the outcome of survival against two different lethal doses (LD) of *Streptococcus pneumoniae*.** Male Oregon-R flies were challenged with a LD₄₀ (dose that kills 40% of flies within 2-5 days post-injection) or a LD₆₀ (dose that kills 60% of flies within 2-5 days post-injection) of *Streptococcus pneumoniae* (*S.p.*) or *S.p.* with 200 ng Hp-FAR-2. Survival was recorded for up to 20 days post-injection and CFU (colony-forming units) quantification was measured at two time points: immediately after injection (0 hour) and 24 hours post-injection (24 hours). **A)** Co-injection of flies with a LD₄₀ of *S.p.* with 200 ng Hp-FAR-2 exhibits no significant outcome of survival compared to the LD₄₀ of *S.p.* flies alone. **B)** No significant change in microbial load was observed at 24 hours post-injection for LD₄₀ *S.p.* plus 200 ng Hp-FAR-2 or LD₄₀ *S.p.* alone. **C)** Co-injection of flies with a LD₆₀ of *S.p.* with 200 ng Hp-FAR-2 exhibits no significant outcome of survival compared to the LD₆₀ of *S.p.* flies alone. **D)** No significant change in microbial load was observed at 24 hours post-injection for LD₆₀ *S.p.* plus 200 ng Hp-FAR-2 or the LD₆₀ *S.p.* alone. Survival experiments were triplicated, with a total of 180 flies per treatment group. Survival data shown as a Kaplan-Meier survival curve and statistical analysis was performed using the Log-rank test. CFU experiments were performed in triplicate, with eight flies per replicate, totaling 24 flies. CFU data shown as a scatter plot, with individual points representing CFU per fly. Statistical significance was analyzed using an unpaired t-test with a line at the median.

controls were used to assess potential background effects arising from GAL4 expression or UAS-mediated activation in the absence of GAL4, respectively. The driver control consisted of TubP-GAL4 flies crossed with wildtype w[1118], termed as TubP>+, while the UAS control consisted of the UAS line crossed with wildtype w[1118], termed as Ch2-UAS>+ or Ch3-UAS>+. Adult male flies ubiquitously expressing Hp-FAR-2 and their respective controls were injected with 6,000 CFU of

*S. pneumoniae*, and survival was monitored for twenty days. No significant difference in survival was observed between male Hp-FAR-2-expressing flies and UAS control flies for either insertion (Fig 5A). However, TubP > Ch2-UAS male flies exhibited a decrease in survival compared to only the driver control (Fig 5A). Furthermore, no significant difference in microbial load was observed between Tub > Ch2-UAS flies and the respective controls (Fig 5B). However, male TubP > Ch3-UAS flies exhibited a significant reduction in microbial load 24 hours post-injection compared to the UAS control Ch3-UAS/+ (Fig 5B).

To examine sex-based differences, we also analyzed the infection outcome in female transgenic flies. No significant difference in survival was observed between female Hp-FAR-2-expressing flies and UAS control flies for either insertion (Fig 5C). Similar to the males, TubP > Ch2-UAS females displayed decreased survival compared to only the driver control.

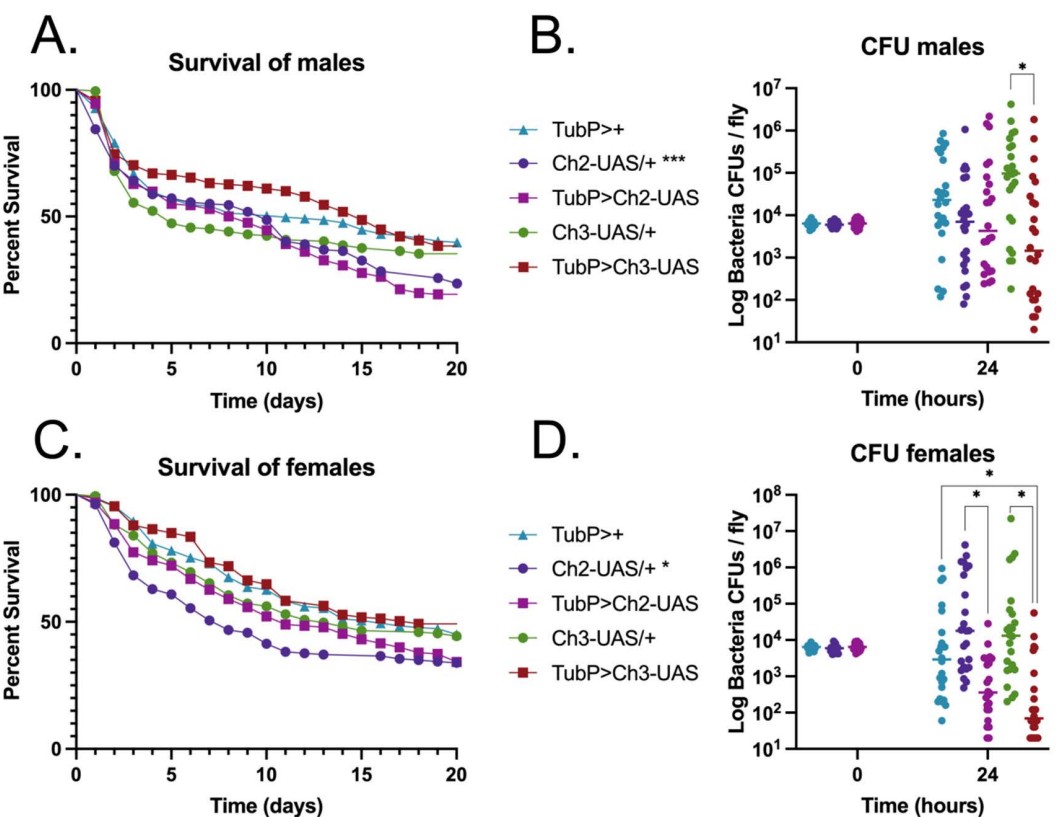

**Fig 5. Analysis of transgenic Hp-FAR-2 expression on survival of male and female flies against *Streptococcus pneumoniae* infection.** Adult male and female flies ubiquitously expressing Hp-FAR-2 using the GAL4/UAS system, along with their respective controls were challenged with 6,000 cells of *Streptococcus pneumoniae*. Survival was recorded for up to 20 days post-injection and colony forming units (CFU) was measured immediately after injection (0 hour) and 24 hours post-injection (24-hours). **A)** Adult male flies ubiquitously expressing Hp-FAR-2 in both chromosomal insertions, TubP > Ch2-UAS and TubP > Ch3-UAS exhibited no significant infection outcome compared to the respective UAS controls (Ch2-UAS/+ and Ch3-UAS/+). TubP > Ch2-UAS males exhibited decreased survival compared to only the driver control TubP>+, indicated in asterisks. **B)** A significant decrease in 24-hour microbial load was observed in chromosome 3 UAS-Hp-FAR-2 male flies (TubP > Ch3-UAS) compared to the UAS control. **C)** Adult female flies expressing Hp-FAR-2 in both chromosomal insertions showed no significant difference in survival compared to their UAS controls. TubP > Ch2-UAS females exhibited decreased survival compared to only the driver control TubP>+. **D)** Adult female flies expressing Hp-FAR-2 exhibited a decrease in microbial load 24 hours post-injection compared to the respective UAS controls and the driver control. Survival experiments were triplicated, with a total of 180 flies per treatment group. Survival data shown as a Kaplan-Meier survival curve and statistical analysis was performed using the Log-rank test. CFU experiments were performed in triplicate, with eight flies per replicate, totaling 24 flies. CFU data shown as a scatter plot, with individual points representing CFU per fly. Statistical significance was analyzed using an unpaired t-test with a line at the median. Asterisks indicate the following p-value cut offs: * $p < .05$.

However, because this effect was observed with only a single control and insertion line, it is insufficient to conclude definitively that expression of Hp-FAR-2 influences survival. Interestingly, Hp-FAR-2 expression significantly reduced microbial load 24 hours post-injection compared to the UAS-control in both insertions, and the driver control for the chromosome three insertion (Fig 5D). Collectively, these results indicate that injection of Hp-FAR-2 does not affect the survival of male and female flies following bacterial infection, and Hp-FAR-2-expressing flies may promote tolerance by reducing microbial load without improving survival. The $w^{1118}$ genetic background of control flies may have contributed to increased microbial load at 24 hours post-injection in males and females, as $w^{1118}$ flies are more susceptible to various bacterial pathogens and have less capacity to induce stronger innate immune activities at early time points during certain bacterial infections [78,80]. These results differ from previous findings on a FAR from *C. elegans* and *A. ceylanicum*, which were found to dampen fly immunity by decreasing resistance to infection [18], suggesting that Hp-FAR-2 may be targeting a host-specific pathway that is not conserved in *Drosophila*.

### Hp-FAR-2 does not affect immune responses to bacterial infection in *Drosophila melanogaster*

To further evaluate the immunomodulatory effects of Hp-FAR-2, we assessed various indicators of immune response to bacterial infection in *D. melanogaster*, such as antimicrobial peptide (AMP) expression, phenoloxidase (PO) activity, and phagocytic activity of hemocytes. First, to evaluate the role of Hp-FAR-2 in humoral immunity, we measured expression of Toll and Imd-dependent AMPs in *D. melanogaster*. Male Oregon-R flies were injected with *S.p.* or *S.p* plus 200 ng recombinant Hp-FAR-2. Injection with PBS alone was tested as a control. The expression of two Imd-dependent AMPs, *defensin* and *diptericin* and two Toll-dependent AMPs, *drosomycin* and *metchnikowin* were measured 18 hours post-injection using quantitative PCR (qPCR). Treatments with *S.p.* resulted in significant upregulation of endogenous *defensin*, *drosomycin,* and *metchnikowin* compared to PBS alone (Fig 6A). Flies coinjected with 200 ng Hp-FAR-2 and *S.p.* showed no significant change to AMP expression compared to injection with *S.p.* alone (Fig 5A). These results indicate that Hp-FAR-2 does not affect the Toll and Imd pathway in response to bacterial challenge of *S. pneumoniae* in *D. melanogaster*.

In addition to AMP expression, we evaluated changes in PO activity in the hemolymph using the DOPA assay. PO is a key enzyme in the *Drosophila* melanization cascade, an immediate immune response to injury and infection. Melanization links humoral and cellular immune response, ultimately resulting in the localized production of melanin at the site of injury or infection to contain the pathogen and facilitate wound healing [81]. This process is mediated by crystal cells, a class of hemocytes that synthesize and release PO into the hemolymph near the injury site. PO then catalyzes the oxidation of phenols to quinones which polymerize into melanin [82]. In this assay, PO catalyzes the conversion of levodopa (L-DOPA) substrate into a pigment called dopachrome (orange to red), whose absorbance can be measured at 492 nm. Male Oregon-R flies were injected with 1,000 colony forming units of Gram-positive intracellular pathogen *Listeria monocytogenes* (*L.m.*), known to mount a robust disseminated melanization response in *D. melanogaster* [83]. Fly hemolymph was collected 6 hours later for PO quantification. We found that injection with *L. monocytogenes* significantly increased PO activity compared to PBS alone (Fig 6B). To measure if Hp-FAR-2 affected PO activity, we coinjected *L. monocytogenes* with 200 ng Hp-FAR-2 and found co-injection with Hp-FAR-2 did not result in significant changes in PO activity compared to *L. monocytogenes* injection alone (Fig 5B). These results indicate that a one-time treatment with 200 ng Hp-FAR-2 had no effect on PO activity in *D. melanogaster*.

Phagocytosis is a crucial cellular immune process mediated by plasmatocytes in *Drosophila* [84]. Phagocytic activity of both Gram-negative and Gram-positive bacteria was quantified through the injection of fluorescently labeled conjugates of *Escherichia coli* and *Staphylococcus aureus* respectively, which fluoresce upon exposure to the low pH environment of the lysozyme. The total fluorescent area, corresponding to successful phagocytosis of the bacteria, was quantified one hour post-injection using image-processing software ImageJ [65]. To measure whether Hp-FAR-2 affected phagocytosis in vivo we co-injected male Oregon-R with conjugated *E. coli* and *S. aureus* at a concentration of 1 mg/mL with either 150 ng Hp-FAR-2 or PBS (control flies). Flies were dissected and imaged one hour post-injection, as phagocytosis of bacteria has

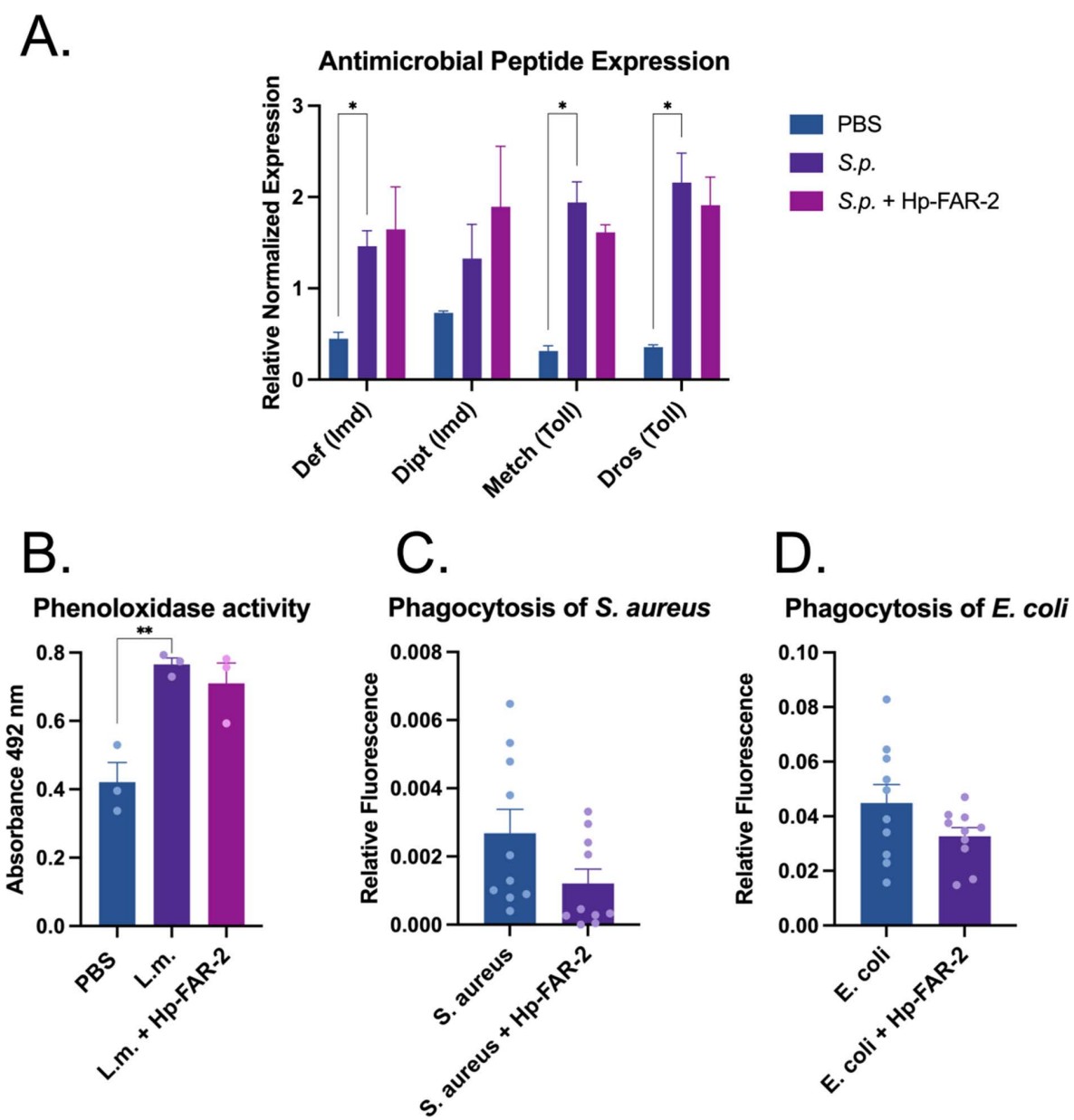

**Fig 6. Various immune pathways in *D. melanogaster* are unaffected by Hp-FAR-2.** Several immune responses were tested in *D. melanogaster* to analyze the immunomodulatory effects of Hp-FAR-2. **A)** Antimicrobial peptide (AMP) expression was measured 18 hours post-injection with *Streptococcus pneumoniae* (*S.p.*), *S.p.* plus 200 ng Hp-FAR-2, or PBS-injected flies. Four different AMPs from the Imd or Toll pathway were measured*, defensin* (Imd), *diptericin* (Imd), *drosomycin* (Toll), and *metchnikowin* (Toll) using qPCR. *Defensin, metchnikowin*, and *drosomycin* were significantly upregulated after injection with *S.p.* compared to PBS. alone. *S.p.* plus 200 ng FAR showed no change in AMP expression relative to *S.p.* alone. Experiments were biologically triplicated with 15 flies per biological replicate. Data shown as bar graphs with error bars depicting the mean with SEM (standard error of the mean). Statistics shown as two-way ANOVA with Dunnett's multiple comparisons test. Asterisks indicate the following p-value cut offs: * $p < .05.$ **B**) Phenoloxidase (PO) activity was measured 6 hours post-injection with either PBS (control), 1,000 cells *Listeria monocytogenes* (*L.m.*), a known melanizer, or *L.m.* plus 200 ng Hp-FAR-2. No significant change in PO activity was observed in *L.m.* plus 200 ng Hp-FAR-2 injected groups. PO was significantly increased in flies injected with *L. m.* compared to PBS injection alone. Experiments were biologically triplicated with 50 flies per biological replicate. Data shown as bar graphs with individual points representing biological replicates. Error bars depict mean with SEM. Statistics shown as an ordinary one-way ANOVA with Dunnett's multiple comparisons test done in GraphPad Prism. Asterisks indicate the following p-value cut offs: ** $p < .01.$ **C)** Phagocytic activity of gram-positive bacteria *S. aureus* was measured with the pHrodo Red *S. aureus* BioParticles assay. No significant change in phagocytosis was observed in flies injected with 150 ng Hp-FAR-2 plus *S. aureus* compared to *S. aureus* plus PBS (control). **D)** Phagocytosis of gram-negative

bacteria *E. coli* was measured with the pHrodo Red *S. aureus* BioParticles assay. Phagocytosis was unaffected by the addition of 150 ng Hp-FAR-2 compared to *E.coli* and PBS (control). All phagocytosis experiments were biologically triplicated with at least 3 flies per biological replicate. Data shown as bar graphs with individual points representing all experimented flies. Statistics displayed as an unpaired t-test with error bars depicting mean with SEM.

been shown to occur in less than 30 minutes in adult flies [85]. Our findings demonstrated that 150 ng Hp-FAR-2 did not affect the phagocytic activity of *S. aureus*, as indicated by no change in phagocytosis compared to *S. aureus* with PBS alone (Fig 6C). No changes in relative fluorescence were observed during experiments with *E. coli* plus 150 ng Hp-FAR-2 compared to *E. coli* with PBS alone, suggesting that Hp-FAR-2 did not affect the phagocytosis of *E. coli* (Fig 6D). Collectively, these results suggest that Hp-FAR-2 does not modulate key aspects of immunity against bacterial challenge in *D. melanogaster*.

### Hp-FAR-2 downregulated M1 and M2 macrophage gene expression in RAW 264.7 cells

We next sought to investigate the functional role of Hp-FAR-2 in host immunity using a more biologically relevant model. A mouse macrophage cell line (RAW 264.7, TIB-71) was utilized to assess its effects on macrophage polarization and phagocytosis. Macrophages are critical to host immunity against *H. polygyrus,* during both primary and secondary infection as the dominant cell type in Type 2 granulomas and first responder to the tissue dwelling larvae [46]. To investigate the effect of Hp-FAR-2 on bacterial phagocytosis, RAW 264.7 cells were pretreated with 1 µg/mL of Hp-FAR-2 or PBS for 24 hours, followed by the addition of pHrodo red-labeled *E. coli* to the cell culture. The cells were fixed, counterstained with DAPI, and subsequently imaged for analysis. Phagocytosis efficiency was measured as the mean fluorescence intensity using ImageJ software. Both PBS- and Hp-FAR-2-treated macrophages showed a strong red fluorescence signal, indicating engulfed bacteria, with blue labeling of the cell nuclei from DAPI (S4A Fig). Our results show that Hp-FAR-2 did not significantly affect macrophage phagocytosis of *E. coli* compared to the PBS control (S4B Fig), suggesting Hp-FAR-2 does not affect phagocytosis of bacteria in RAW 264.7 macrophages.

To further investigate if Hp-FAR-2 can influence host macrophage function, RAW 264.7 cells were polarized into classically activated M1 or alternatively activated M2 macrophages using IFN-γ and LPS or IL-4 respectively, with non-stimulated (M0) macrophages included as a control. During secondary infection, CD4 + T cell-dependent Th2-type memory is required for host protection against the parasite, which includes the induction and recruitment of AAMacs by IL-4/IL-13 [44,46,86]. During infection, these AAMacs have been found to highly express a variety of immunoregulatory and wound healing molecules including chitinase-like proteins that have lost their enzymatic activities, e.g., Chil3, resistin-like molecules including RELM- α, and Arg1 [46]. To determine whether Hp-FAR-2 affected macrophage polarization, cells were co-cultured with 3 µg/mL Hp-FAR-2 or PBS. Macrophage polarization was determined by measuring gene expression of *TNF-α* and *IL-6* for M1 type, and *Chil3* and *Arg1*, for M2 type. Hp-FAR-2 exerts a potent immunomodulatory effect on macrophage polarization. In M1-like macrophages, Hp-FAR-2 significantly modulated marker gene expression, reducing expression of M1 markers *TNF-α* and *IL-6* compared to the PBS treated controls (Fig 7A). When helminth parasites infect their mammalian host, epithelial and endothelial barriers are damaged, which induces a wound repair and an anti-parasite immune response that is driven by the type 2 cytokines IL-4, IL-5, and IL-13 [87]. Effective wound repair requires both the direct reconstruction of the injured tissue and the suppression of pro-inflammatory responses [87]. It is hypothesized that helminths have evolved strategies to suppress inflammation, which may serve both their own survival and contribute to host tissue homeostasis. We hypothesize that Hp-FAR-2 suppresses expression of *TNF-α* and *IL-6* in M1 macrophages to limit inflammation, as classical Th1-mediated responses can cause extensive host tissue damage. Similar to our findings, *H. polygyrus* secreted EVs have been found to suppress macrophage activation in BMDMs resulting in a downregulation of *IL-6*, *TNF-α*, and *Chil3* [88]. A previous study found that secretion of both TNF-α and IL-6 was reduced in LPS-stimulated RAW 264.7 cells pretreated with DHA [89]. Similar results were found with other PUFAs such as LA, which

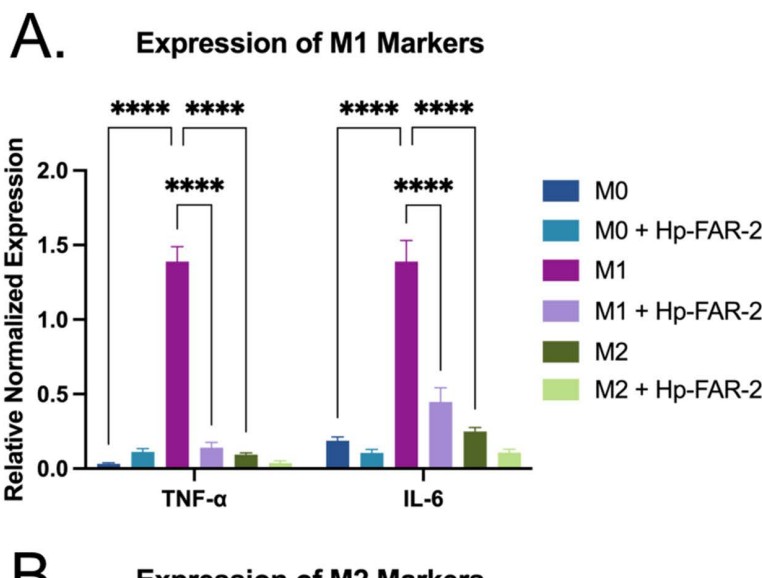

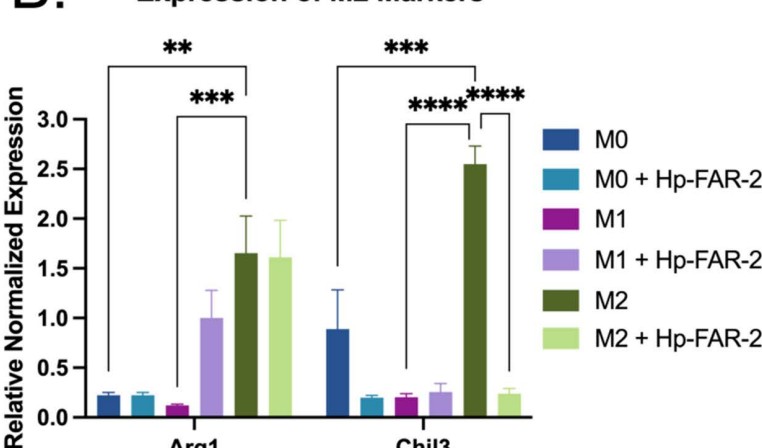

**Fig 7. Exposure to Hp-FAR-2 dampens expression of M1 and M2 markers in RAW 264.7 macrophages.** RAW 264.7 macrophages were cultured with PBS (M0), INF-γ/LPS (M1), or IL-4 (M2), with the addition of and 3 µg/mL Hp-FAR-2 or PBS as a control for 48 hours. Expression of M1 and M2 markers were measured using qPCR. **A)** Expression of M1 markers *TNF-α* and *IL-6* was significantly downregulated in Hp-FAR-2 treated cells compared to the control (M1). **B)** Quantification of M2 marker expression revealed that *Arg1* expression was not significantly altered by Hp-FAR-2 treatment compared to the M2 control, whereas Hp-FAR-2 significantly reduced *Chil3* expression compared to M2 cells. No significant changes in M1 or M2 markers were observed in unpolarized macrophages (M0) with or without Hp-FAR-2. The experiment was carried out with three biological replicates per treatment, with plots shown as bar graphs and error bars depicting mean with SEM. Statistics shown as a two-way ANOVA with Tukey's multiple comparisons test. Asterisks indicate the following p-value cut offs: ** $p < 0.01$, *** $p < 0.001$, **** $p < 0.0001$.

decreased the production of TNF-α and IL-6 in RAW264.7 macrophages exposed to LPS [90]. Furthermore, OA was found to inhibit LPS-induced TNF-α secretion in mouse monocytes [91]. Therefore, it is unlikely that sequestration of DHA, LA, or OA by Hp-FAR-2 would decrease expression of *TNF-α* and *IL-6* in INF-γ/LPS activated macrophages, suggesting that Hp-FAR-2 may be binding to other lipids or retinol. Hp-FAR-2 was found to competitively bind, in ten-fold excess, to a variety of lipids including omega-3 and omega-6 PUFAs, a monounsaturated fatty acid, and a saturated fatty acid (Fig 3A). We hypothesize that Hp-FAR-2 may be sequestering other lipids, such as PA, to attenuate expression of M1 markers in INF-γ/LPS activated macrophages. Studies have shown that PA can increase TNF-α and IL-6 production in mouse primary

macrophages without the presence of LPS [92], and can increase TNF-α secretion in LPS-induced mouse monocytes [91].

Similarly, in M2 macrophages, Hp-FAR-2 significantly reduced expression of *Chil3* but had no effect on *Arg1* expression (Fig 7B). Furthermore, no significant changes in M1 or M2 markers were observed in unpolarized macrophages (M0) with or without Hp-FAR-2 (Fig 7). This suggests that Hp-FAR-2 may function to suppress *Chil3* expression in M2 macrophages, potentially limiting excessive tissue repair that could otherwise be detrimental to parasite survival. Arg1 is thought to function in controlling inflammation and tissue repair during *H. polygyrus* infection. Arginase metabolism can result in increased polyamine production which is known to down-regulate Th1-type inflammation [93]. Macrophages expressing high levels of *Arg-1* have been implicated in the deposition of collagen and formation of fibrotic granulomas [94]. Finally, an arginase-1 product, L-ornithine, was found to directly impair *H. polygyrus* larval motility [95]. Chil3 binds to heparin on cell surfaces and in the extracellular matrix [96], suggesting it may contribute to repair of tissue damage caused by *H. polygyrus* when migrating through the intestinal wall [97]. Furthermore, reduced *Chil3* expression may promote increased susceptibility to *H. polygyrus* survival in the host, as a more resistant mouse strain, SJL, was found to develop macrophage-rich granulomas at the intestinal site of invasion, with high levels of Chil3 protein both in and around the granuloma [98]. Our data show that Hp-FAR-2 exhibits a competitive binding preference for DHA (Fig 3), and previous data have shown a correlation between DHA and macrophage plasticity and polarization phenotype in mice [99]. Bone marrow-derived (BMDM) M2 macrophages from DHA-deficient mice showed significant downregulation in the expression of M2 marker *STAT6* with no effect on *Arg1* expression; these results were restored in cells obtained from DHA-supplemented animals [99]. Although this study did not analyze the effects of DHA on *Chil3* expression in M2 macrophages, many studies have identified the essential role of STAT6 in IL-4 signaling and *Chil3* expression in AAMacs [44,100]. We hypothesize that Hp-FAR-2 may be sequestering DHA locally, hindering expression of *Chil3*, potentially mediated through a STAT6-dependent mechanism, while leaving *Arg1* expression in M2 macrophages unaffected. Further research is needed to elucidate a mechanistic link between *Chil3* expression and Hp-FAR-2 treatment in the context of M2 macrophage polarization in mice. Overall, modulating inflammation and promoting a Th2 response is likely advantageous for *H. polygyrus*, as it helps minimize immune-mediated damage, enhance tissue repair, and establish a more tolerant environment conducive to chronic infection [44,46,101].

More research is needed to elucidate the mechanism by which Hp-FAR-2 modulates gene expression, including testing a broader range of doses and using flow cytometry to further understand the cell-specific effects on macrophage polarization. One limitation to this study is the absence of a control protein, which should be included in future studies to confirm the specificity of the observed effects. It is important to note the relevance of the dosage of Hp-FAR-2 treatment in the context of nematode infections. Estimates show that 200 L3 larvae yield 150–200 adult worms in mice, each producing ~20 ng of the HES daily—totaling 6–8 µg of ES products over 48 hours [53]. We used a total of 1.2 µg Hp-FAR-2 in our 48-hour polarization assays, which is well below the calculated physiologically relevant dose of the HES. It is important to note that different mouse strains may vary in susceptibility to infection leading to variations of ES product concentrations per strain.

In summary, this study shows that Hp-FAR-2 does not modulate immunity or survival in *D. melanogaster*, unlike FARs from *C. elegans*, *S. carpocapsae*, and the human hookworm *A. ceylanicum*, suggesting functional divergence within this protein family. This distinction highlights the specificity and diversity among different FARs, indicating that individual FARs may have evolved unique roles in host-parasite interactions that are not conserved across different hosts. Our findings demonstrate a potential role for Hp-FAR-2 in mediating omega-3 and omega-6 fatty acid uptake in chronic intestinal infection by *H. polygyrus.* Furthermore, we found that Hp-FAR-2 can dampen expression of M1 and M2 markers in RAW 264.7 cells during polarization, suggesting it may act as a broad immunomodulator, specifically targeting macrophages, which play an important role in clearing parasitic infection. The identification of Hp-FAR-2 in the HES during different life stages provides further evidence that FAR proteins play a crucial role in nematode parasitism. A deeper understanding of FARs role in parasitic nematode infections is essential for elucidating host-parasite interactions and may contribute to the development of more effective anthelmintic therapies and novel treatments for autoimmune diseases.

## Supporting information

**S1 Fig.  Size-exclusion chromatography (SEC) and SDS-PAGE analysis of recombinant Hp-FAR-2. A)** SEC elution profile of Hp-FAR-2 fraction. Using SEC, Hp-FAR-2 was found to contain on primary peak, highlighted in green, and the fractions collected for further analysis are indicated by the red line. **B)** SDS-PAGE analysis of the collected SEC fractions. The elution fractions (A8, A10, A12, B7, B5, B3, B1) are loaded alongside a molecular weight marker in (left lane) with molecular weights labeled in kDa. A prominent band around 17 kDa (shown under the red line) indicates the presence of the purified protein.
(PDF)

**S2 Fig.  Sequence analysis of Hp-FAR-2.** A consensus casein kinase II phosphorylation site at residue 46–49 (under-lined), predicted signal peptide (highlighted in grey), and predicted cleavage site (red) are shown.
(PDF)

**S3 Fig.  Hp-FAR-2 is not toxic to *Drosophila melanogaster.*** Flies were injected with PBS, or 200 ng Hp-FAR-2 and survival was recorded for 20 days post-injection. Hp-FAR-2 injected flies show no significant difference in survival from PBS-injected flies. Survival is graphed as a Kaplan-Meir with log-rank test p value significance indicated by an asterisk.
(PDF)

**S4 Fig.  Hp-FAR-2 does not affect phagocytic activity of RAW 264.7 macrophages.** Cells were incubated with Hp-FAR-2 (1 µg/ml) or PBS for 24 hours prior to exposure to pHrodo-labeled *E. coli*. Cells were fixed and counterstained with DAPI and subsequently imaged for analysis. **A)** Representative images show RAW 264.7 cells stained with DAPI (blue) and pHrodo-labeled *E. coli* (red), with merged channels displayed. Scale bar = 50 µm. **B)** Phagocytic activity was quantified as mean fluorescence intensity using ImageJ, with ~1250 cells analyzed per replicate and 4 biological repli-cates per treatment. Data are presented as mean + SEM. Statistical analysis was performed using an unpaired t-test in GraphPad Prism.
(PDF)

**S1 Table.  List of fatty acid and retinol binding proteins (FARs) in the *H. polygrus* genome.** FAR proteins were extracted from the *H. polygrus* genome assembly project (NCBI accession: PRJEB15396) and (NCBI accession: PRJEB1203) with WormBase ParaSite Ensembl BioMart tool, filtering for the Gp-FAR-1 domain (Pfam: 05823). The table includes gene ID, project ID, and protein name if applicable, with project PRJEB15396 and respective FAR genes labeled in red.
(PDF)

**S2 Table.  Sequences of primers used in all qPCR reactions.**
(PDF)

**S3 Table.  Mass spectrometry data from transgenic *Drosophila* expressing Hp-FAR-2.**
(PDF)

**S1 Data.  Macro used in ImageJ for image processing for in vivo *Drosophila* pHrodo assay.**
(PDF)

## Acknowledgments

We would like to thank Brittany Anabenawa-Appiah for assistance with fly husbandry and Kyle Anesko for aid with mouse macrophages. We would also like to thank Quanqing Zhang at the UCR Proteomics Core for assisting with mass spectrometry.

## Author contributions

**Conceptualization:** Pakeeza Azizpor, Adler R. Dillman.

**Data curation:** Pakeeza Azizpor.

**Formal analysis:** Pakeeza Azizpor.

**Funding acquisition:** Adler R. Dillman.

**Investigation:** Pakeeza Azizpor, Janice Montoya, Fayez Eyabi, Jose Ramirez, Tara Hill, Robert Pena, Manisha Mishra.

**Methodology:** Pakeeza Azizpor, Robert Pena, Manisha Mishra, Martin J. Boulanger.

**Project administration:** Adler R. Dillman.

**Resources:** Adler R. Dillman.

**Supervision:** Martin J. Boulanger, Adler R. Dillman.

**Validation:** Pakeeza Azizpor.

**Visualization:** Pakeeza Azizpor.

**Writing – original draft:** Pakeeza Azizpor, Adler R. Dillman.

**Writing – review & editing:** Pakeeza Azizpor, Manisha Mishra, Martin J. Boulanger, Adler R. Dillman.

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
