## [Decision Letter · Decision Letter 0]

9 Jul 2025

PNTD-D-25-00888

A secreted fatty acid- and retinol- binding protein from Heligmosomoides polygyrus suppresses host macrophage polarization

Dear Dr. Dillman,

Thank you for submitting your manuscript to PLOS Neglected Tropical Diseases. After careful consideration, we feel that it has merit but does not fully meet PLOS Neglected Tropical Diseases's publication criteria as it currently stands. Therefore, we invite you to submit a revised version of the manuscript that addresses the points raised during the review process.

Please submit your revised manuscript within 60 days Sep 07 2025 11:59PM. If you will need more time than this to complete your revisions, please reply to this message or contact the journal office at plosntds@plos.org. Please include the following items when submitting your revised manuscript:

We look forward to receiving your revised manuscript.

Kind regards,

Ramesh Ratnappan, PhD

Academic Editor

Jong-Yil Chai

Section Editor

Shaden Kamhawi

co-Editor-in-Chief

Paul Brindley

co-Editor-in-Chief

**Journal Requirements:**

At this stage, the following Authors/Authors require contributions: Pakeeza Azizpor, Janice Montoya, Fayez Eyabi, Jose Ramirez, Tara Hill, Robert Pena, Manisha Mishra, Martin Boulanger, and Adler R. Dillman. Please ensure that the full contributions of each author are acknowledged in the "Add/Edit/Remove Authors" section of our submission form.

3) We noticed that you used the phrase 'data not shown' in the manuscript. We do not allow these references, as the PLOS data access policy requires that all data be either published with the manuscript or made available in a publicly accessible database. Please amend the supplementary material to include the referenced data or remove the references.

4) We do not publish any copyright or trademark symbols that usually accompany proprietary names, eg ©,  ®, or TM  (e.g. next to drug or reagent names). Therefore please remove all instances of trademark/copyright symbols throughout the text, including:

- ® on pages: 6, and 8

- TM on pages: 6, 7, 8, 9, 10, and 20.

5) We have noticed that you have uploaded Supporting Information files, but you have not included a list of legends. Please add a full list of legends for your Supporting Information files after the references list.

6) Thank you for stating 'All data is available in the supplemental and through Mendeley Data via this link: https://data.mendeley.com/preview/2bt7xj2ydj?a=91cf1766-8422-4085-8e3e-d41b368d4e25'. We found that this link directs to a 'dataset not found' page. Kindly fix it to point to an active link.

7) Please amend your detailed Financial Disclosure statement. This is published with the article. It must therefore be completed in full sentences and contain the exact wording you wish to be published.

2) If any authors received a salary from any of your funders, please state which authors and which funders..

8) Please send a completed 'Competing Interests' statement, including any COIs declared by your co-authors. If you have no competing interests to declare, please state "The authors have declared that no competing interests exist". Otherwise please declare all competing interests beginning with the statement "I have read the journal's policy and the authors of this manuscript have the following competing interests:"

**Reviewers' Comments:**

Reviewer's Responses to Questions

**Key Review Criteria Required for Acceptance?**

**Methods**

-Are the objectives of the study clearly articulated with a clear testable hypothesis stated?

-Is the study design appropriate to address the stated objectives?

-Is the population clearly described and appropriate for the hypothesis being tested?

-Is the sample size sufficient to ensure adequate power to address the hypothesis being tested?

-Were correct statistical analysis used to support conclusions?

-Are there concerns about ethical or regulatory requirements being met?

Reviewer #1: (No Response)

Reviewer #2: The objectives of the study are articulated along with a clear hypotheses of addressing the protein Hp-FAR-2 and it's functional divergence from other FAR proteins in drosophila model. I could not find information regarding the number of data points studied in the manuscript itself. I would suggest including a data section with information about the sample size.

Reviewer #3: A well focused study on a single fatty acid binding protein (Hp-FAR-2) from a widely used model helminth system, H. polygyrus, with a sophisticated Drosophila transgenesis system and some data from mouse macrophages.

Statistically all sound except Figure 5 A and C there is clearly a protective effect of Hp-FAR-2 on fly survival that perhaps an overly stringent statistical test is classifying as not significant.

Reviewer #4: The methods are appropriate to the subject of the study. Very little effect was found of HpFAR2 expression in Drosophila, but I have concerns about the experiments using recombinant protein on RAW264.7 cells:

- The protein was expressed in E coli, and there is no mention of removing the signal peptide, which would not be removed by expression in a prokaryotic system. It would have been better to express the protein without the signal peptide to give the functional secreted protein.

- more importantly, the protein was expressed in E coli, and it was not stated if endotoxin levels were measured. E coli expressed proteins are very often purified with contaminating endotoxin, and as RAW264.7 cells are highly sensitive to endotoxin stimulation, it could well be that the effects of HpFAR2 addition are due to endotoxin, rather than the protein itself. These experiments should be repeated with protein which has been shown to be endotoxin free, and a suitable control protein expressed in the same system should also be compared.

**Results**

-Does the analysis presented match the analysis plan?

-Are the results clearly and completely presented?

-Are the figures (Tables, Images) of sufficient quality for clarity?

Reviewer #1: (No Response)

Reviewer #2: The analysis presented matches the analysis plan, although I would urge the authors to improve clarity in language for the analysis plan. Figure 7(a) for E. coli isn't sufficiently clear for publication.

Reviewer #3: The results are presented in reasonable fashion but several figures could be more clearly annotated and there are a number of questions and deficiencies.

(1) In Figure 1, the coloring is confusing, indicating greater identify than actually is the case, some errors (eg KE at 36-37 not colored), and starkly different residues given same color (eg Phe to Ala or Ile at 56/57). All substitutions have a functional significance. Better to show identities only.

(2) Line 446 and elsewhere, the mass of 19.5968 kDa is (a) false accuracy and (b) refers to the full length protein including the signal peptide that does not exist outside the protein synthesis machinery.

(3) Line 453 and elsewhere 91.11% identity is another example of false accuracy. The amino acid identity is over 180 residues so each difference counts for 0.55%, ie one less identity and the difference would be 90.55%, so no more than 3 significant figures are appropriate.

(4) Figure 2 should include a negative control protein to show the background level of nonspecific DAUDA binding.

(5) Figure 3 shows 1:1 and 1:10 ratios, but this is quite coarse grained, is there any reason not to do a full titration to better quantify the “binding preference”? The Figure should spell out names of the fatty acids tested.

(6) Also clarify annotations in Figure 5, "TunP>Ch2-UAS" takes quite some deciphering, and colors on the line graph are too similar to discern.

(7) In general not clear how doses were selected, for example only a single injection of 200 ng of Hp-FAR-2 was tested for effects over 20 days; macrophages were cultured with 3 ug/ml.

(8) It is not clear whether key data on macrophages (Figure 8) represents repeated experiments, ie were the “biological replicates” all in the same experiment, or was this result obtained from ≥2 independent experiments, one of which is represented in the Figure? The Legend does not specify timing, there is no control protein added, and the measures are restricted to PCR for mRNA (and hence does not warrant the term "marker" which implies measurement of cell surface moieties by flow cytometry).

(9)

Reviewer #4: Data presented in an appropriate manner

**Conclusions**

-Are the conclusions supported by the data presented?

-Are the limitations of analysis clearly described?

-Do the authors discuss how these data can be helpful to advance our understanding of the topic under study?

-Is public health relevance addressed?

Reviewer #1: (No Response)

Reviewer #2: Limitations of analysis such as the doses of hp-FAR-2 are addressed in the manuscript. Authors also address the role Hp-FAR-2 may play in lipid absorption and modulate expression of other lipids.

Reviewer #3: In general, the conclusions are matched by the data, but the key finding (quoted in the title) that the protein "suppresses host macrophage polarization" is not well supported, with a single experiment (Figure 8) at a single dose using only RT-PCR and no control protein. This should be repeated and extended over a dose range and coupled with flow cytometry.

Reviewer #4: Throughout the manuscript, including the abstract, it is stated that Omega3/6 fatty acids are immunoactive, however little evidence is presented for this. The reference for Oleic acid (the target with the best evidence for binding) being immunoactive is a single review article, and is not conclusive.

In the discussion, it is suggested that the lipid binding ability of HpFAR2 could switch host cells to production of immunoactive lipids such as the prostaglandins, leukotrienes etc. However these are lal arachadonic acid metabolites, which HpFAR2 does not seem to preferentially bind to. This should be discussed.

**Editorial and Data Presentation Modifications?**

Reviewer #1: (No Response)

Reviewer #2: Minor revisions needed. The manuscript tries to cover too much ground, making it difficult to identify the key findings and relate the data to findings. Other revisions mentioned above.

Reviewer #3: The manuscript is very, very wordy, with the Discussion repeating much of the results; a substantial reduction in length would be appropriate, as would moving some of the negative data (eg figure 7) to supplementary information.

Reviewer #4: Line 61-65: this sentence is unclear (double negative) and should be rewritten.

**Summary and General Comments**

Reviewer #1: General comments

The paper deals with a family of proteins from nematodes, the FAR proteins, that are being increasingly recognised for their importance in parasitic nematodes controlling the immune defence reactions of their hosts. Such effects mediated by FAR proteins have been found in nematode infections of plants and animals. Understanding how these proteins have their effects is potentially important for developing means by which the health of humans, domestic animals, and crop plants could be improved. The paper aims to examine the effect of a FAR protein (Hp-FAR-2) from a parasitic nematode of mice, Heligmosomoides polygyrus, a parasite with well-described immunomodulatory properties. But most of the experiments reported examine the influence of the protein in Drosophila melanogaster insects rather than in a mammalian system. This approach is imaginative but assumes that the FAR protein concerned will operate similarly across phyla of hosts, even though it has similar lipid-binding properties to most other FAR proteins so far described. The authors have previously shown that a FAR from an insect parasite does have effects on Drosophila, which could be because that protein engages in interactions beyond the binding of lipids. In the present paper, however, the Hp-FAR-2 protein is shown to have no effects on immune system parameters in flies or on their survivability in bacterial infections. The authors then jump to examining the effect of Hp-FAR-2 on macrophages from mice in the search for some effects. The paper therefore reads like two separate papers – one that shows no effects in flies, then one that shows some effects on a completely different host system – the latter being on the face of it a more straightforward (though with much room for development) examination of the effects of a protein from a parasite of mice on an aspect of the murine immune system. This juxtaposition does seem confusing.

Specific points.

1. The title does not properly represent the paper’s content. As said above, the bulk of the paper deals with Hp-FAR-2 in Drosophila and that the mouse macrophage aspect is tacked on the end, yet the title mentions only the latter.

2. Line 70 – citation [34] refers to a FAR protein (entirely appropriate to the paper) but not to a different class of proteins from nematodes as intended. I did a search and see that the same group as mentioned in that citation published on the polyprotein type - doi.org/10.1371/journal.pntd.0001040 , which is perhaps what is intended?

3. Line 274 – Is this statement true for all FAR proteins?

4. Line 464 – What is “the FAR protein domain”?

5. Figure 1B – If included, it would be useful to indicate the cleavable secretory signal peptides in the alignment.

6.Figures 2A and 2C – The spectra look remarkably noise-free – have they been smoothed? Also, in Materials and Methods it is stated that the spectra are corrected for background fluorescence of PBS, presumably by subtraction, but the signal for PBS is included and is higher in places than the spectra with DAUDA +/- Hp-FAR-2, presumably because it is the uncorrected version? Would it not be more straightforward merely to show all the spectra uncorrected? Incidentally, the signal for PBS alone is quite high, particularly at the shorter wavelengths – is there a reason for that?

7. Figure 2B – the fitted curve does not fit the data points well. Is this because of the aberrant last point? If so, might its removal improve the fit? Other than improving the affinity calculation it could improve the statement about relative affinity with retinol.

8. Figure 2D – The curve fit is not ideal for retinol either, and the data points are very scattered. Retinol can be subject to decay in water – has this been compensated for? In lines 483 and 484 it is stated that “These values reveal that relative to retinol, Hp-FAR-2 has a higher affinity for fatty acids.” Care might be needed here because it could be risky to generalise to fatty acids from a calculation based on a fatty acid with quite a large fluorophore attached. Also, given the above comments on the curve fitting, and the intrinsic errors involved, the calculated affinities for DAUDA and retinol are well within an order of magnitude, so may not be meaningfully different.

9. The lipid preferences of the protein show, in Figure 3 reveal a surprise in that Hp-FAR-2 shows little or no affinity for linolenic acid. Is there any precedent for this seeming high degree of specificity in this protein family, and do the authors have any suggestion for how it comes about?

10, Lines 481-482 – “The Kd for Hp-FAR-2:retinol was determined by increasing the concentration of retinol (0-30 μM) in the presence of 1 μM DAUDA” – is this correct?

11. Figure 5 – Understanding the list of experimental groups to the right in the figure requires considerable reference to the figure’s legend and the main text. Can it be simplified to make clear what each experimental group represents?

12. Figure 7A&B – This figure shows that Hp-FAR-2 does not enhance phagocytosis. But might this be a case for adding a positive control protein to demonstrate that the assay is working?

Minor points.

When doing BLAST searches and the like to find similar proteins it is preferable to use the term “similarity” (which is what one searches for and identifies) rather than “homology” (which is an evolutionary inference).

“Data” is plural, so where it occurs it should read “data were”.

“Primary sequence” as used in line 465 – this should be “primary structure” which, for a protein refers to its amino acid sequence. Or just say “sequence”.

The term “technical replicates” is used a few times, but what does it mean – merely a replicate?

Reviewer #2: The research addresses an important problem with clear biomedical relevance. The experimental approach is multi-faceted, using both Drosophila infection models and mammalian cell culture to characterize the protein. The finding of functional divergence within the FAR family is noteworthy and adds to our understanding of these proteins.

Reviewer #3: An incremental study representing a considerable body of work with Drosophila that was not so successful, and a single experiment on mouse macrophages that needs strengthening if it is to be claimed in the Title.

Reviewer #4: This manuscript concerns a fatty acid and retinol binding protein from H. polygyrus. it shows the binding specificity of the protein, and tests it in Drosophila and RAW264.7 macrophage cell line experiments.

It is well presented and written, however the potential caveat of endotoxin contamination of purified protein must be addressed to allow the conclusions to be drawn.

PLOS authors have the option to publish the peer review history of their article (what does this mean? ). If published, this will include your full peer review and any attached files.

**Do you want your identity to be public for this peer review?** For information about this choice, including consent withdrawal, please see our Privacy Policy .

Reviewer #1: No

Reviewer #2: **Yes: ** Nafisa Bulsara

Reviewer #3: No

Reviewer #4: **Yes: ** Henry McSorley

**Figure resubmission:**
---

## [Decision Letter · Decision Letter 1]

29 Sep 2025

Dear Dr. Dillman,

We are pleased to inform you that your manuscript 'Characterization and ligand binding properties of a fatty acid- and retinol- binding protein (Hp-FAR-2) from Heligmosomoides polygyrus' has been provisionally accepted for publication in PLOS Neglected Tropical Diseases.

Best regards,

Ramesh Ratnappan, PhD

Academic Editor

Jong-Yil Chai

Section Editor

Shaden Kamhawi

co-Editor-in-Chief

Paul Brindley

co-Editor-in-Chief

Reviewer's Responses to Questions

**Key Review Criteria Required for Acceptance?**

**Methods**

-Are the objectives of the study clearly articulated with a clear testable hypothesis stated?

-Is the study design appropriate to address the stated objectives?

-Is the population clearly described and appropriate for the hypothesis being tested?

-Is the sample size sufficient to ensure adequate power to address the hypothesis being tested?

-Were correct statistical analysis used to support conclusions?

-Are there concerns about ethical or regulatory requirements being met?

Reviewer #3: (No Response)

Reviewer #4: (No Response)

**Results**

-Does the analysis presented match the analysis plan?

-Are the results clearly and completely presented?

-Are the figures (Tables, Images) of sufficient quality for clarity?

Reviewer #3: (No Response)

Reviewer #4: (No Response)

**Conclusions**

-Are the conclusions supported by the data presented?

-Are the limitations of analysis clearly described?

-Do the authors discuss how these data can be helpful to advance our understanding of the topic under study?

-Is public health relevance addressed?

Reviewer #3: (No Response)

Reviewer #4: (No Response)

**Editorial and Data Presentation Modifications?**

Reviewer #3: (No Response)

Reviewer #4: (No Response)

**Summary and General Comments**

Reviewer #3: The authors have responded to all my comments in a thoughtful and comprehensive manner, amending the title to a more accurate representation of the work presented, and condensing the Discussion, as well as attending to each of the specific points raised, and revising the manuscript accordingly. The revised manuscript is fully suitable for publication.

Reviewer #4: Th authors have addressed my previous comments.

PLOS authors have the option to publish the peer review history of their article (what does this mean? ). If published, this will include your full peer review and any attached files.

**Do you want your identity to be public for this peer review?** For information about this choice, including consent withdrawal, please see our Privacy Policy .

Reviewer #3: No

Reviewer #4: **Yes: ** Henry McSorley

---

## [Editor Report · Acceptance letter]

Dear Dr. Dillman,

We are delighted to inform you that your manuscript, "Characterization and ligand binding properties of a fatty acid- and retinol- binding protein (Hp-FAR-2) from Heligmosomoides polygyrus," has been formally accepted for publication in PLOS Neglected Tropical Diseases.

Best regards,

Shaden Kamhawi

co-Editor-in-Chief

Paul Brindley

co-Editor-in-Chief
